# Retrograde signaling by a mtDNA-encoded non-coding RNA preserves mitochondrial bioenergetics

A. Blumental-Perry [1✉], R. Jobava[2], I. Bederman[2], A. J. Degar[3], H. Kenche[4,14], B. J. Guan[2], K. Pandit[5], N. A. Perry[2], N. D. Molyneaux[2], J. Wu[2], E. Prendergas[2], Z.-W. Ye[6], J. Zhang[6], C. E. Nelson [4], F. Ahangari[7], D. Krokowski[8], S. H. Guttentag[9], P. A. Linden[10,11], D. M. Townsend[12], A. Miron[2], M.-J. Kang [7], N. Kaminski [7], Y. Perry[13✉] & M. Hatzoglou [2✉]

Alveolar epithelial type II (AETII) cells are important for lung epithelium maintenance and function. We demonstrate that AETII cells from mouse lungs exposed to cigarette smoke (CS) increase the levels of the mitochondria-encoded non-coding RNA, mito-RNA-805, generated by the control region of the mitochondrial genome. The protective effects of mito-ncR-805 are associated with positive regulation of mitochondrial energy metabolism, and respiration. Levels of mito-ncR-805 do not relate to steady-state transcription or replication of the mitochondrial genome. Instead, CS-exposure causes the redistribution of mito-ncR-805 from mitochondria to the nucleus, which correlated with the increased expression of nuclear-encoded genes involved in mitochondrial function. These studies reveal an unrecognized mitochondria stress associated retrograde signaling, and put forward the idea that mito-ncRNA-805 represents a subtype of small non coding RNAs that are regulated in a tissue- or cell-type specific manner to protect cells under physiological stress.

[1] Department of Biochemistry, Jacobs School of Medicine and Biomedical Sciences, University at Buffalo, State University of New York, Buffalo, NY, USA. [2] Department of Genetics and Genome Sciences, School of Medicine, Case Western Reserve University, Cleveland, OH, USA. [3] College of Pharmacology, Mercer University, Atlanta, GA, USA. [4] Biomedical Sciences, Mercer University School of Medicine, Savannah Campus, Savannah, GA, USA. [5] Sekusui XenoTech, LLC, Kansas City, KS, USA. [6] Department of Cell and Molecular Pharmacology and Experimental Therapeutics, Medical University of South Carolina, Charleston, SC, USA. [7] Division of Pulmonary, Critical Care and Sleep Medicine, Department of Internal Medicine, and Center for RNA Science and Medicine, Yale School of Medicine, New Haven, CT, USA. [8] Department of Molecular Biology, Maria Curie-Skłodowska University, Lublin, Poland. [9] Division of Neonatology, Vanderbilt University School of Medicine, Nashville, TN, USA. [10] Department of Surgery, Case Western Reserve University School of Medicine, Cleveland, OH, USA. [11] University Hospitals Cleveland Medical Center, Cleveland, OH, USA. [12] College of Pharmacy, Drug Discovery & Biomedical Sciences, Medical University of South Carolina, Charleston, SC, USA. [13] Division of Thoracic Surgery, Department of Surgery, Jacobs School of Medicine and Biomedical Sciences, University at Buffalo, State University of New York, Buffalo, NY, USA. [14]Present address: Savannah State University, Savannah, GA, USA. ✉email: annablum@buffalo.edu; yperry@buffalo.edu; mxh8@case.edu

   **1**

Cigarette smoke (CS) is associated with over 400,000 deaths annually in the United States and is a major risk factor for chronic obstructive pulmonary disease (COPD)[1,2]. COPD is characterized by airspace enlargement due to extracellular matrix destruction and loss of alveolar epithelial Type-I (AETI) cells[3]. AETI cells cover 95% of the alveolar surface, forming a thin barrier for gas exchange[4]. Alveolar epithelial Type-II (AETII) cells produce surfactants, which are crucial for the alveolar structure[5]. These cells are also local progenitors that can proliferate and *trans*-differentiate into AETI cells to maintain lung homeostasis and to repair damage in response to injury[6,7]. CS alters AETII cell metabolism[8–10], and metabolic changes are associated with mitochondrial malfunction.

CS exposure has complex effects on mitochondrial biology, depending on the cell type and on the dose and duration of CS exposure. In many cell types, CS exposure causes fragmentation of mitochondria, a switch to glycolysis as the predominant source of energy, activation of the inflammasome, mitophagy, and if mitochondrial homeostasis is not re-established mitochondria-mediated programmed cell death (apoptosis, necroptosis, ferroptosis, etc.)[10–18]. AETII cells were found to be more resistant to CS-induced injury than AETI cells[19], which is in agreement with their progenitor-like role in lung injury repair. The same doses of CS that cause mitochondria-mediated death in the majority of cell types can induce biogenesis, and stress response-mediated "hormesis"-like effects in AETII cells[20]. In AETII, CS exposure causes mitochondrial elongation[21]. Elongated mitochondria can better maintain energy production under stress[22], therefore elongation is a sign of adaptation.

This resistance suggests of the existence of molecular mechanisms that protect AETII cells from initial CS-induced injury[23]. It is not known whether mitochondrial elongation is the only protective mechanism in AETII cells.

The mammalian mitochondrial genome is a 16.5-kb circular double-stranded DNA molecule, encoding 13 essential protein subunits of electron transport chain (ETC) complexes, two rRNA subunits, and 22 tRNAs[24,25]. Reports identified several classes of non-coding (nc) RNAs produced by the mitochondrial DNA (mtDNA) (mito-ncRNAs). These include different classes of mito-miRs (17–24 nucleotides (nts)), mito-piRNAs (24–31 nts), mito-ncRNAs (<200 nts), and mito-lncRNAs (>200 nts)[26–28]. The functions of mito-ncRNAs are being elucidated. mito-miRs modify host genes in vivo, and their deregulation leads to altered expression of mitochondrial genes[29]. Other classes of mito-ncRNAs have been proposed to be involved in communication between mitochondria and nucleus, with suggested roles during stress responses[30]. The mechanisms of their export from mitochondria and import to the nucleus are not known[31]. Ago2 has been reported in mitochondria, while the presence of DICER and other RNAi machinery has not been identified in the mitochondria proteome[32]. The precise mechanisms of maturation, trafficking through intracellular space, and function of mito-ncRNAs are undetermined[27].

We screened for the CS-induced changes in the expression of microRNAs in AETII cells and identified a number of mircoRNAs whose expression was induced by CS. Characterized here, mito-ncR-805 was found to be not an miRNA but a 70-nt RNA encoded by mtDNA, induced by CS in a cell-type-specific manner and shuttle into the nucleus. Mito-ncR-805 has a positive effect on the expression of subset of nuclear genes, such as subunits of ETC, influences mitochondrial metabolism, and bioenergetics. The function of mito-ncR-805 represents a previously unknown molecular mechanism of retrograde signaling that protects AETII cells from CS-induced injury.

## Results

**CS exposure altered the expression of microRNAs in AETII cells.** To identify changes in microRNAs in response to CS exposure, we used the AETII-like surfactant-producing mouse lung epithelial 12 (MLE12) cells[33,34]. MLE12 cells treated with 10% CS extract (CSE) were assessed for miRNA expression using Agilent mouse miRNA microarrays (Fig. 1a, c, d). Concentrations of CSE were titrated to induce an adaptive chronic stress response; 10 h of CSE treatment had <10% cell loss (Supplementary Fig. 1a). Two-hour CSE treatment induced phosphorylation of *eIF2α*, an indicator of translational repression in the early phase of stress responses[35,36]. Ten hours induced the expression of GADD34 and recovery from stress protein (Fig. 1b)[37]. CSE treatment increased phosphorylation of AKT, a pro-survival kinase, in early and late phases, as described previously[38]. Ten hours of CSE exposure also led to a reduction in TRIB3 protein, a known inhibitor of AKT phosphorylation, which fostered sustained P-AKT during the recovery phase (Fig. 1b)[35,39]. Principal component analysis of miRNA expression showed that analyzed groups are statistically unique, with the expression pattern at 10 h of CSE exposure more similar to that of the control cells than the 2 h, supporting that 10-h CSE exposure corresponds to the recovery phase at the level of known stress-signaling components and miRNA expression (Fig. 1c).

Hierarchical clustering demonstrated that out of 627 miRNAs analyzed, 19 are downregulated and 7 are upregulated (Fig. 1d and Supplementary Data). CSE exposure has been demonstrated to affect Dicer function in some cell types, leading to a global non-specific downregulation of miRNA expression[40]. We did not observe global downregulation of all miRNAs in MLE12 cells but considered the upregulated miRNAs as potential candidates for specific CSE-induced changes, focusing on miRNAs increased at 10 h of CSE exposure as potential mediators of recovery. The miRNAs validated to meet these criteria were miR-805, with the highest fold induction (Fig. 1e, f), miR-709, and miR-1907 (Supplementary Fig. 1b, c)[41–43].

The upregulation of miR-805 was validated in isolated primary mouse AETII cells (Fig. 1g) exposed to CSE ex vivo using adjusted concentrations and exposure times (Fig. 1h). Increased levels of miR-805 were also observed in primary AETII cells isolated from 3-month CS-exposed mice (Fig. 1i). Therefore, miR-805 is induced in response to CSE in MLE12 and primary AETII cells ex vivo and in vivo.

We tested whether induction of miR-805 is a general response of different cell types. miR-805 levels were compared in total lung and liver lysates of control and CS-exposed mice. The levels of miR-805 were downregulated in total lung CS-exposed samples (Supplementary Fig. 1d, e). Liver is a tissue that shares common properties with AETII cells: secretory cells with strong reparative abilities. Expression of miR-805 was elevated in the livers of CS-exposed mice (Supplementary Fig. 1f). Therefore, increase in miR-805 expression in response to CS exposure in mice maybe specific to secretory and local niche progenitor cells.

**miR-805 is an mtDNA-encoded ncRNA, not an microRNA.** Sequence analysis showed that miR-805 maps to mtDNA (Fig. 2a)[44]. Because mitochondria are severely affected by CSE[9–14], we sought to investigate the regulation of miR-805 in the mitochondrial response to CSE.

The mitochondrial localization of miR-805 was supported by enrichment of miR-805 in mitochondrial fractions compared to cytosolic fractions and total cell lysates (Fig. 2b, c). Mitochondrial miR-805 was resistant to RNase digestion, similar to Cox3 and opposite of glyceraldehyde 3-phosphate dehydrogenase (GAPDH) transcripts (Supplementary Fig. 2a). miR-805 maps to

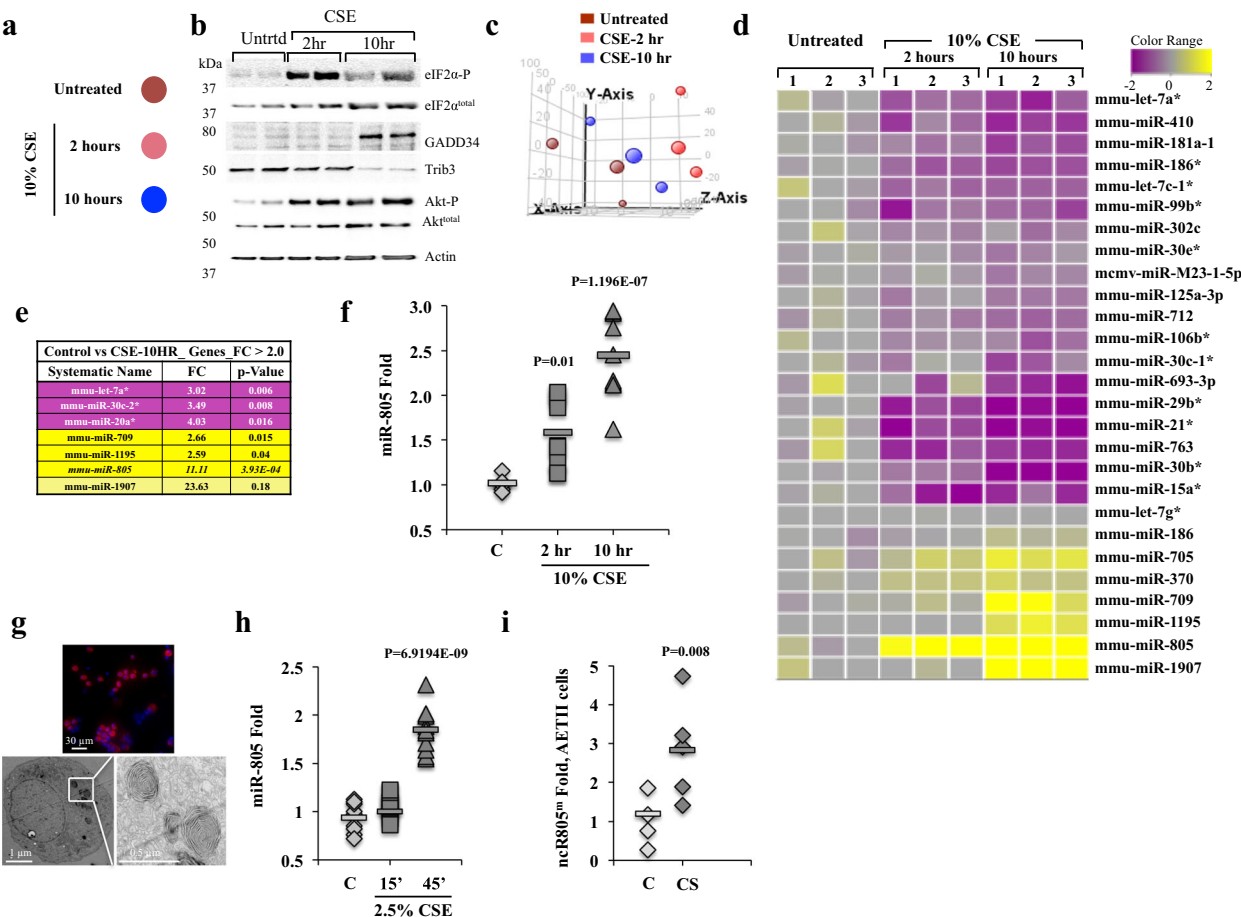

**Fig. 1 Identification of miRNAs regulated by CSE treatment. a** Key of experimental treatments of MLE12 cells with 10% CSE for 0 (untreated), 2, or 10 h. **b** Untreated and treated with 10% CSE for 2 and 10 h MLE12 cells were assayed for changes in protein levels associated with stress signaling. Actin was used as a loading control. **c** GeneSpring 11.0 principle component analysis of untreated cells, and two time points of CSE treatment. **d** Hierarchical clustering of 27 differentially expressed miRNAs obtained from three independent biological experiments. Expression of microRNAs was assayed in untreated and 10% CSE-exposed MLE12 cells using GeneSpring GX 11.0. **e** Table of select microRNAs from **d** with the highest changes. **f** RT-qPCR was used to confirm miR-805 expression in MLE12 cells exposed to 10% CSE for 2 h ($n = 3$ independent experiments/with 9 independent samples, $p = 0.01$) and 10 h ($n = 3$, independent experiments/with 9 independent samples, $p = 1.196E{-}07$). **g** Primary AETII cells were isolated from mouse lungs and their phenotype was verified using antibodies specific to staining lamellar bodies (3C9), showing a 92% purity of isolated AETII cells (111 3C9-positive cells out of total of 121 DAPI nuclei, upper panel). Isolated AETII cells were plated, allowed to adhere and recover from the isolation procedure for 3 days, fixed using Karnovsky's fixation protocol, and processed for TEM to visualize laminar bodies. Images were acquired using JEM 1011 TEM microscope. **h** Mouse primary AETII cells from four independent isolations were plated, cultured for 3 days, and exposed to 2.5% CSE for 15 and 45 min. miRNA-enriched RNA was isolated and tested for the expression of miR-805 by RT-qPCR (Control $n = 3$ independent experiments/with 9 independent samples, 15 min $n = 3$ independent experiments/with 9 independent samples, $p$ value $= 0.006$, and 45 min $n = 3$ independent experiments/with 12 independent samples, $p$ value $= 06.9194E{-}09$). **i** Primary AETII cells isolated from mice exposed to CS for 3 months twice daily ($n = 5$ animals for control, and $n = 5$ animals for CS-exposed animals, $p = 0.008$). All $p$ values indicate the comparison of treated sample values to respective control untreated. RT-qPCR levels of mito-ncR-805 were normalized to sno55RNA in all panels; folds calculated to respective controls.

the light-strand promoter (LSP) D-loop regulatory region of the mtDNA, a major non-coding region involved in the regulation of transcription and replication (Fig. 2d)[24]. The sequence of miR-805 has no other homology within the mitochondria genome. The predicted mature sequence of miR-805 has no homology anywhere within the nuclear genome, thus providing an additional evidence that miR-805 is of a mitochondrial origin.

miR-805 is a predicted miRNA, not tested experimentally in cells. The 5′-end of the predicted pre-miR maps before the previously established initiation site of LS transcription (Fig. 2d)[45]. We performed Northern blot analysis of RNA from CSE-exposed cells, as well as from Ago2- and Dicer-depleted cells (Fig. 2e and Supplementary Fig. 2b, c). Analysis demonstrated that the predicted miR-805 exists within an approximately 70-bp ncRNA, with no smaller than 70 bp RNA species. The levels of the 70 bp

RNA species did not change in Ago2- or Dicer-depleted cells (Fig. 2e and Supplementary Fig. 2d).

To determine the precise length and sequence of the ncRNA, strand-specific RNA libraries of RNAs ranging in length between 15 and 100 nts were constructed and sequenced from control and CSE-exposed MLE12 cells (Supplementary Fig. 2e). The length and sequence of the transcripts were identified by comparison of sequence reads to mouse mitochondrial sequence coordinates 16,210–16,116. This mapping revealed an RNA sequence of 70 bp that corresponds to coordinates 16,188–16,119 of mtDNA (Fig. 2a, f), which is in agreement with the size revealed by Northern blot analysis (Fig. 2e). RNA from CSE-exposed cells exhibited more reads of this 70 bp sequence than control cells (Fig. 2g). The mFold RNA folding program of the predicted pre-miR-805 secondary structure (Supplementary Fig. 2f) requires

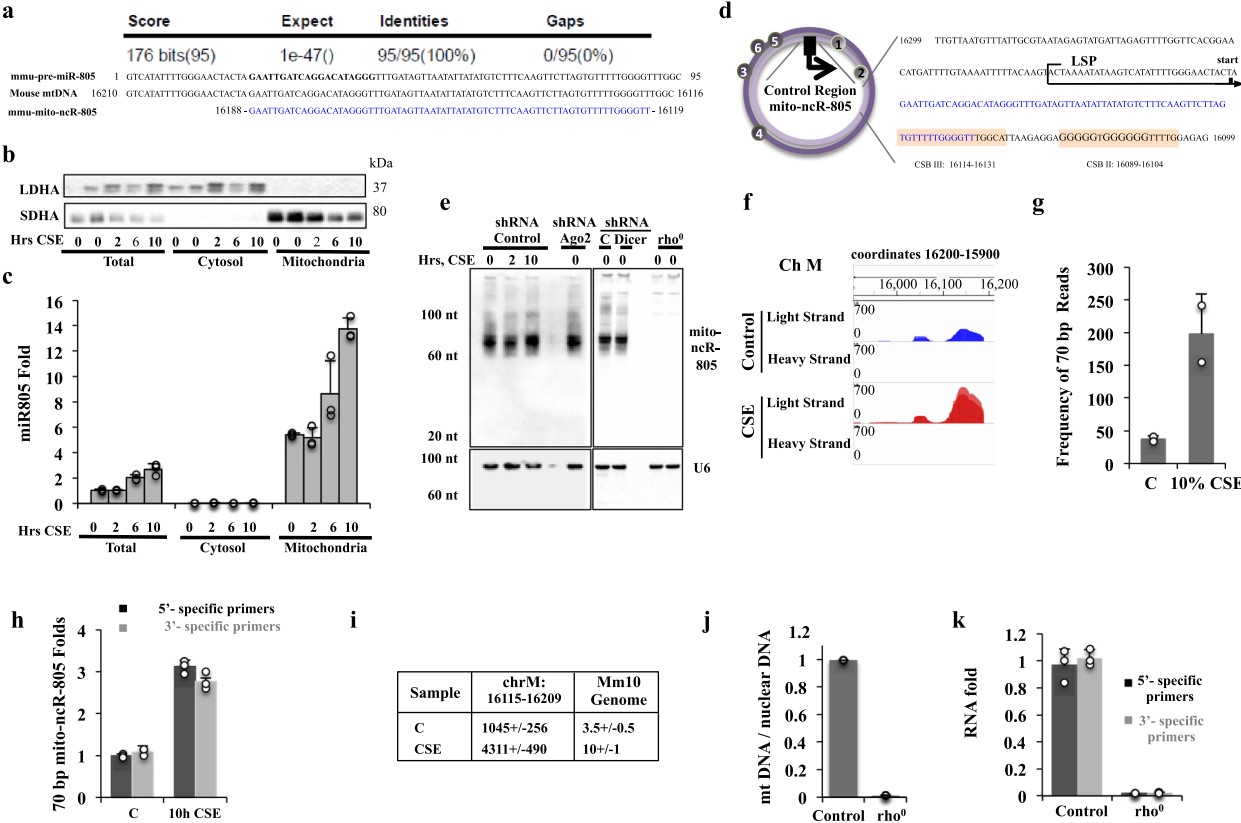

**Fig. 2 miR-805 is an mtDNA-encoded non-coding RNA. a** Alignment of the predicted miR-805 to the mouse mitochondrial genome. The last row depicts the sequence obtained by RNA-sequencing analysis. **b**, **c** MLE12 cells were exposed or not to 10% CSE; cytosolic and mitochondrial extracts were generated. Fractions were analyzed for **b** cytosolic protein lactate dehydrogenase A (LDHA) and mitochondrial protein succinate dehydrogenase subunit A (SDHA) and **c** the expression levels of miR-805. **d** Schematic representation of the mito-ncR-805 genomic location. The circular mtDNA with the heavy (H) strand in dark purple, the light (L) strand in light purple, and the LSP indicated by the black arrow. A portion of the mtDNA control region near the LSP is shown with the H-strand nucleotide sequence. The LSP transcription initiation start site is indicated. The 5′-end of mito-ncR-805 (blue) maps one nucleotide downstream of the LSP transcription initiation site and the 3′-end maps within the conserved sequence block (CSB) III (orange box). mito-ncR-805 appears to be a product of the LSP promoter transcription. CSB II with a G-quartet important for transcription pausing and R-loop formation is also indicated (orange box). **e** Control (non-targeting), Ago2-, and Dicer-specific shRNA MLE12 cells, as well as rho[0] MLE12 cells were treated with 10% CSE for the indicated times. Small RNA-enriched fractions were resolved on a 15% urea-gel and hybridized with probes complementary to miR-805 and to U6 (loading control). **f** Transcription profile of mitochondrial transcripts generated from 16,000–16,210 region of mtDNA. **g** Summary of frequency of reads of the 70-bp transcript. **h** Expression levels of mito-ncR-805 using two sets of primers, directed against the 5′ end (first 20 bp of mito-ncR-805), or directed against the 3′-end of mito-ncR-805 (30–55 bp of mito-ncR-805). **i** Alignment of all the sequence reads against the entire mouse nuclear and mitochondrial genomes. **j–k** rho[0] MLE12 cells were obtained by growing cells in the presence of ethidium bromide added fresh daily for 2 passages. **i** DNA was extracted and mitochondrial copy number was determined (Control 100%, rho[0] 4%). **k** mito-ncR-805 expression levels in MLE12 control and rho[0] cells.

base paring with a non-transcribed part of the mitochondria genome, because the LS transcription start site is located at 16,189[45]. RNAseq confirmed that no transcript is produced from any sequence prior to bp 16,188 (Fig. 2a, d, f), indicating that bp 16,188 is the LS transcription initiation site and the 5′ end of the RNA. This 70 bp transcript was almost undetectable by quantitative PCR (qPCR) in cytosolic fractions (Fig. 2c), indicating that this ncRNA does not require the conventional processing machinery. The CSE-mediated fold induction was similar when measured with probes generated against either the 5′-end or the 3′-end of the 70-nt RNA, further supporting a non-miRNA nature of the transcript (Fig. 2h). The specificity of read alignments to the mitochondrial genome was confirmed by aligning all the sequence reads against the entire mouse nuclear and mitochondrial genomes (Fig. 2i and Supplementary Fig. 2g). The first 29 bp of the 70-bp transcript were found to be unique for the mitochondrial genome, with no homology elsewhere in the nuclear genome; the rest of the transcript has two short homology stretches to two nuclear genes (Supplementary Fig. 2h).

These data strongly support miR-805 being a mito-ncRNA that maps at the LS transcription initiation site. Henceforth, we refer to it as mito-ncR-805.

To unequivocally confirm the mitochondrial origin of mito-ncR-805, we tested its expression levels in MLE12 rho[0] cells[27], in which mtDNA was depleted by ~96% (Fig. 2j)[46]. Depletion of mtDNA resulted in a dramatic decrease in the ability to detect mito-ncR-805 by qPCR (Fig. 2k) and disappearance of mito-ncR-805-specific band from the Northern blot (Fig. 2e).

**mito-ncR-805 distribution in cells**. To visualize mito-ncR-805, a BaseScope probe was developed. This probe detected a dispersed granular network, with concentration in the peri-nuclear region, which is in agreement with known mitochondrial distribution (Fig. 3a). mito-ncR-805-specific staining pattern was completely abolished in rho[0] MLE12 cells (Fig. 3a). Therefore, the ncRNA associated with CS exposure is produced from the mtDNA, enriched in the mitochondrial fraction treated with RNase, and has no homology in the nuclear genome.

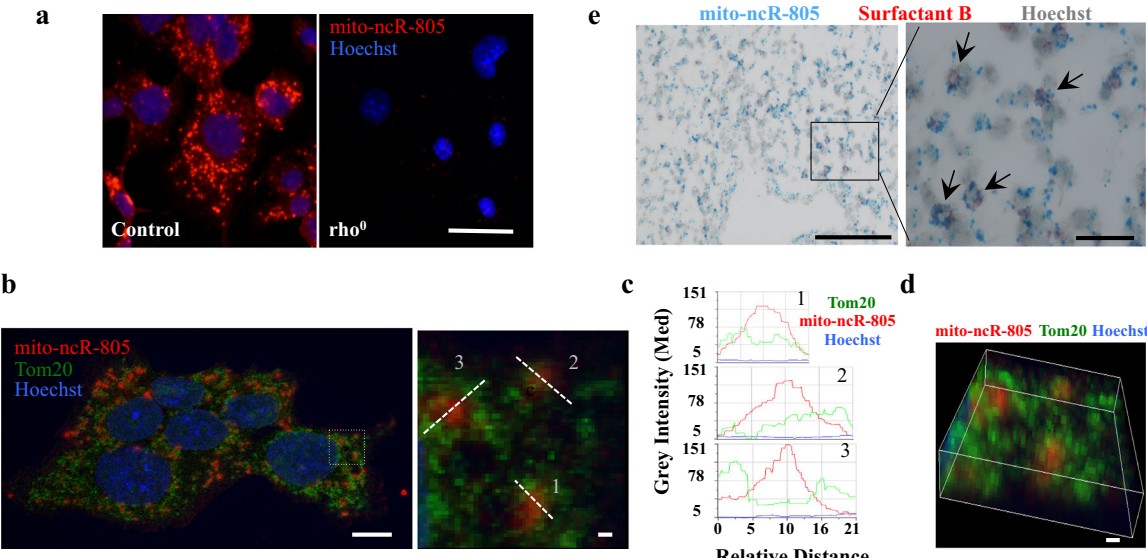

**Fig. 3 mito-ncR-805 cellular distribution. a** Control and rho[0] MLE12 cells were grown on slides, fixed, and hybridized with a mito-ncR-805-specific probe. Images were acquired using ×40 magnification on Leica DM IRE2. Scale bar is 20 μm. **b–d** MLE12 cells were grown on slides, fixed, and hybridized with mito-ncR-805-specific probe, followed by Tom20 staining. Confocal images were acquired using ×63 magnification and HuVolution function of the Leica SP8-HyVolution laser scanning confocal microscope. **b** A representative image, which is a projection of 14 consecutive planes acquired at the distance of 0.13 μm. Scale bar is 10 μm; boxed region on the left is shown enlarged on the right, as a single plane chosen for its maximal signal intensity for each channel. Linescans from the analysis of mito-ncR-805 and Tom20 relative cellular localizations are shown. **c** Representative examples of linescans and plots of three channel intensities through depicted regions; each represents a specific mito-ncR-805 and Tom20 spatial relationship: 1—complete overlap, 2—partial overlap, 3—engulfment of mito-ncR-805 by Tom20. Scale bar is 1 μm. **d** 3D reconstitution of 14 consecutive planes of boxed in **b** area. Scale bar is 1 μm. Images in **b–d** are representatives of three independent biological experiments. For each experiment, 3–7 linescans per cell were analyzed for n = 6 cells **e** Lungs from control mice were inflated and paraffin embedded. mito-ncR-805 (blue dots) and AETII cells, positive for Surfactant B (red) expression, were visualized by ISH. Images were acquired at a magnification of ×20. Scale bar is 100 μm. Boxed area is enlarged; image was acquired at a magnification of ×100. Scale bar is 20 μm. Arrows point to the AETII cells.

Co-detection of mito-ncR-805 and the mitochondrial outer membrane protein Tom20 was performed using fluorescence in situ hybridization–immunofluorescence (FISH-IF) and revealed partial co-localization (Fig. 3b–d), with varying degrees of overlap in signals. Linescans were employed to analyze the spatial relationship between mito-ncR-805 and Tom20. Three predominantly equal relationships were detected: complete, partial co-localizations, and mito-ncR-805 being "engulfed" by Tom20 (Fig. 3c and Supplementary Fig. 2j). mito-ncR-805 dots vary in size significantly, with a single visible dot being 0.25–2 μm in diameter (Fig. 3b). Differences in mito-ncR-805 dot size could not be accounted for by amplification steps during FISH; it is likely that larger dots may represent accumulation of several mito-ncR-805 transcripts. Dots smaller in diameter likely co-localize with Tom20, and dots >1 μm are likely representing mito-ncR-805-containing complexes, which remain in the vicinity of Tom20. Three-dimensional (3D) reconstitution of multiple planes corroborates this possibility (Fig. 3d) and supports the idea of mito-ncR-805 accumulating in some kind of mitochondrial granules.

To test mito-ncR-805 distribution in the lung tissues, BaseScope-duplex ISH was performed using the same probe as was used in MLE12 cells to detect mito-ncR-805 and a probe specific to Surfactant B mRNA to identify AETII cells. Figure 3e demonstrates that mito-ncR-805 is expressed ubiquitously through different cell types in the lung parenchyma. Levels of its expression were scored using modification of a 4-tier scoring system previously used for semi-quantitative microscopical evaluation of RNAScope: 0 = no spots, 1 = few spots, 2 = moderate (<10) spots, 3 = high (>10) spots per cell)[47]. The average number of mito-ncR-805 spots per cell was 10.7 +/− 3.9

per AETII cell (score = 3), and 2.15 +/− 0.4 spots per non-AETII cell (score = 1, Supplementary Data).

In conclusion, mito-ncR-805 is a mitochondrial ncRNA, stored in granular form, and likely present in greater amounts in AETII cells.

**mito-ncR-805 appears in the nucleus during CSE exposure.** To study the dynamics of mito-ncR-805 during the stress of chronic CSE exposure, cells were exposed to 6% CSE for 10 h (induces adaptive response with no associated cell death). mito-ncR-805 localization with respect to Tom20 and the nucleus was visualized via FISH-IF (Fig. 4a). Co-localization was analyzed as a function of overlapping signal intensities obtained using 3D reconstruction of all cellular planes (Fig. 4b), where I reflect mito-ncR-805 relationship to Tom20, and II to nucleus, and linescan analysis (Fig. 4c). Starting at 1 h of CSE exposure, overlap between mito-ncR-805 and Tom20 signals increase (Fig. 4b, c), which is in agreement with new synthesis of mito-ncR-805 in the mito-chondria or its exit from mitochondria. Nuclear localization of mito-ncR-805 becomes visible at 3 h of CSE exposure, as dis-tinctive red spots in the nucleus (Fig. 4b), and it gradually increases at 6 h (Fig. 4b, c), reaching its maximum at 10 h, where the whole nucleus appears purple from mito-ncR-805 signal (Fig. 4b, c).

As nuclear localization increases, the number of mito-ncR-805 dots decreases (Fig. 4d), with 10 h exhibiting very few. Levels of mito-ncR-805 from the same experiment demonstrated the expected increase in mito-ncR-805 expression levels by qPCR (Supplementary Fig. 3a).

To confirm the presence of mito-ncR-805 in the nucleus, total and nuclear extracts with no observed mitochondria

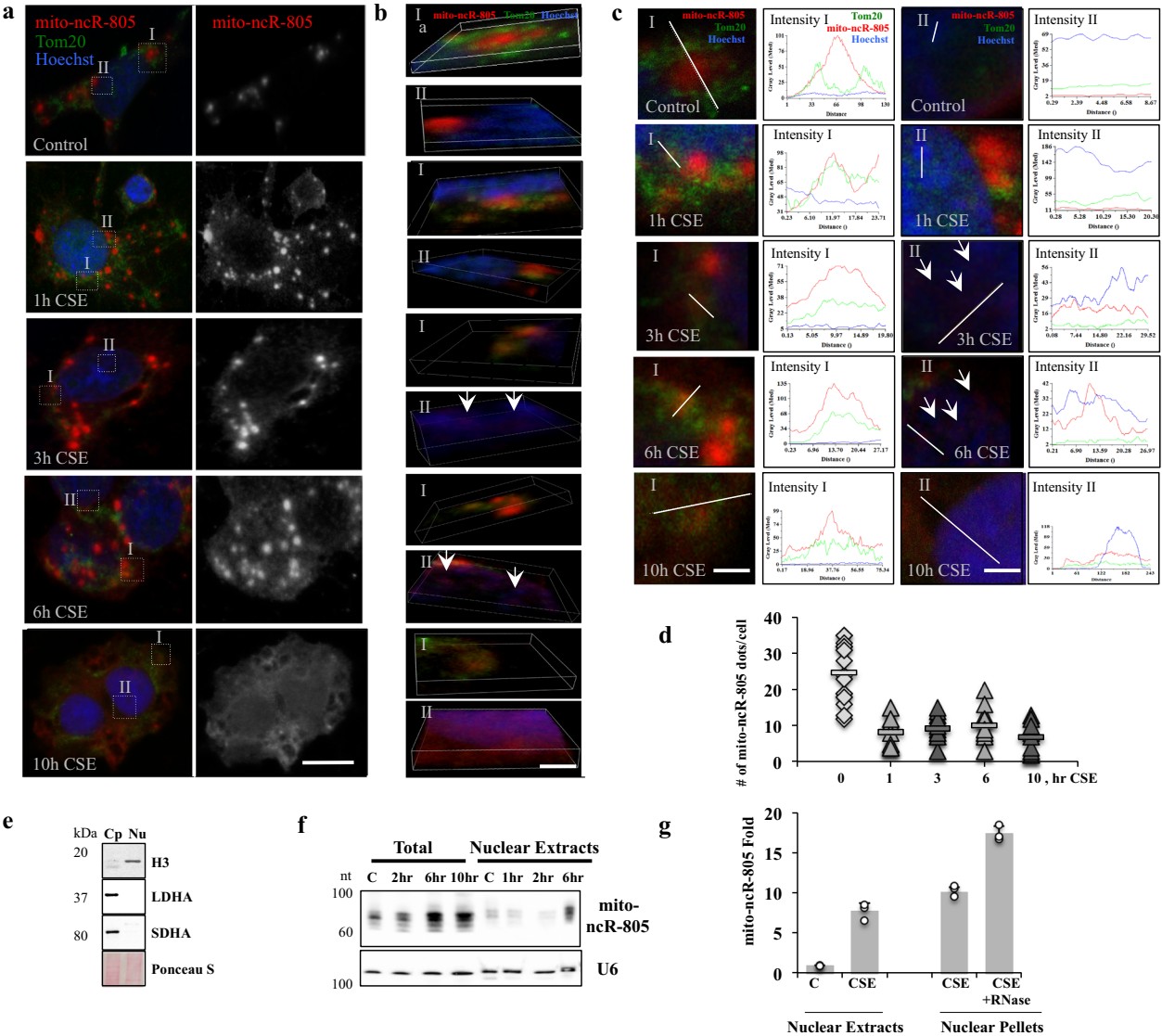

**Fig. 4 mito-ncR-805 redistributes from mitochondria and appears in the nucleus during exposure of MLE12 cells to CSE.** MLE12 cells treated with 6% CSE for the indicated times were analyzed for stress-associated changes in mito-ncR-805 subcellular distribution. **a** Subcellular distribution of mito-ncR-805 and Tom20 was evaluated using FISH-IF. Nuclei were visualized by Hoechst. Scale bar is 20 μm. Boxed areas I are enlarged representative examples of areas analyzed for mito-ncR-805 and Tom20 localization. Boxed areas II are enlarged representative examples of areas analyzed for mito-ncR-805 localization in relationship to the nucleus. Confocal images were acquired using ×63 magnification. A representative image is a projection of 7 consecutive planes acquired at the distance of 0.3 μm. **b** 3D reconstitution of representative boxed areas from **a**. Scale bar is 2 μm. Arrows at the 3 h CSE time point to the initial mito-ncR-805 spots in the nucleus (**c**). Linescan analysis of areas I and II with representative examples of plots of three channel intensities through regions depicted by linescans. Images in **a**–**c** are representatives of three independent biological experiments. For each experiment, 3–4 cells for each condition were analyzed, with 3–5 linescans per each cell. **d** Quantification of mito-ncR-805 dots per individual cell counted for 12–15 cells for each time point. p values: $p^{Control/1h} = 1.96458E{-}05$, $p^{Control/3h} = 3.78809E{-}07$, $p^{Control/6h} = 2.73901E{-}06$, $p^{Control/10h} = 0.000686789$; not statistically significant between different exposure time: $p^{1h/10h} = 0.543742736$. **e, f** MLE12 cells were exposed to CSE for 0, 2, 6, and 10 h. Total, nuclear, and cytoplasmic extracts were prepared. **e** Purity of nuclear extracts (Nu), as compared to cytoplasmic extracts (Cp). Proteins were isolated from half of each fraction, and fractions were analyzed for cytosolic protein LDHA, mitochondrial protein SDHA, and nuclear protein Histone 3 (H3). **f** Small RNAs were isolated from the other half of total and nuclear fraction, resolved on 15% urea-gel, and probed with the probes complimentary to mito-ncR-805 and to U6. **g** Levels of mito-ncR-805 were analyzed by RT-qPCR in total and nuclear extracts and in nuclear pellets treated with RNase.

contamination (Fig. 4e) were prepared from MLE12 cells exposed or not to CSE. The small amount of mito-ncR-805 in the nucleus was detectable in Control cells and the amount increased several fold in CSE-exposed cells. The increase in the nucleus coincided with the recovery phase (6–10 h) when a rise in the total levels of mito-ncR-805 is also observed (Fig. 4f). Nuclear mito-ncR-805 was resistant to RNAse treatment (Fig. 4g).

We found no decrease in mtDNA, and mitochondrial morphology was intact, with no visible mitochondrial dispersion

or interaction with the nucleus (Supplementary Fig. 3b–e). CSE exposure of MLE12 cells did not cause decrease in any intermediates of tricarboxylic acid (TCA) cycle but resulted in trending increase in the activity of the TCA cycle but not glycolysis (Supplementary Fig. 3f–g). Therefore, mitochondria of MLE12 cells survive CSE exposure and remain active, providing a support to the idea that the discovered phenomenon is a part of an adaptive stress response. The dispersion of the in situ signal is specific to mito-ncR-805, not mitochondria. ISH and

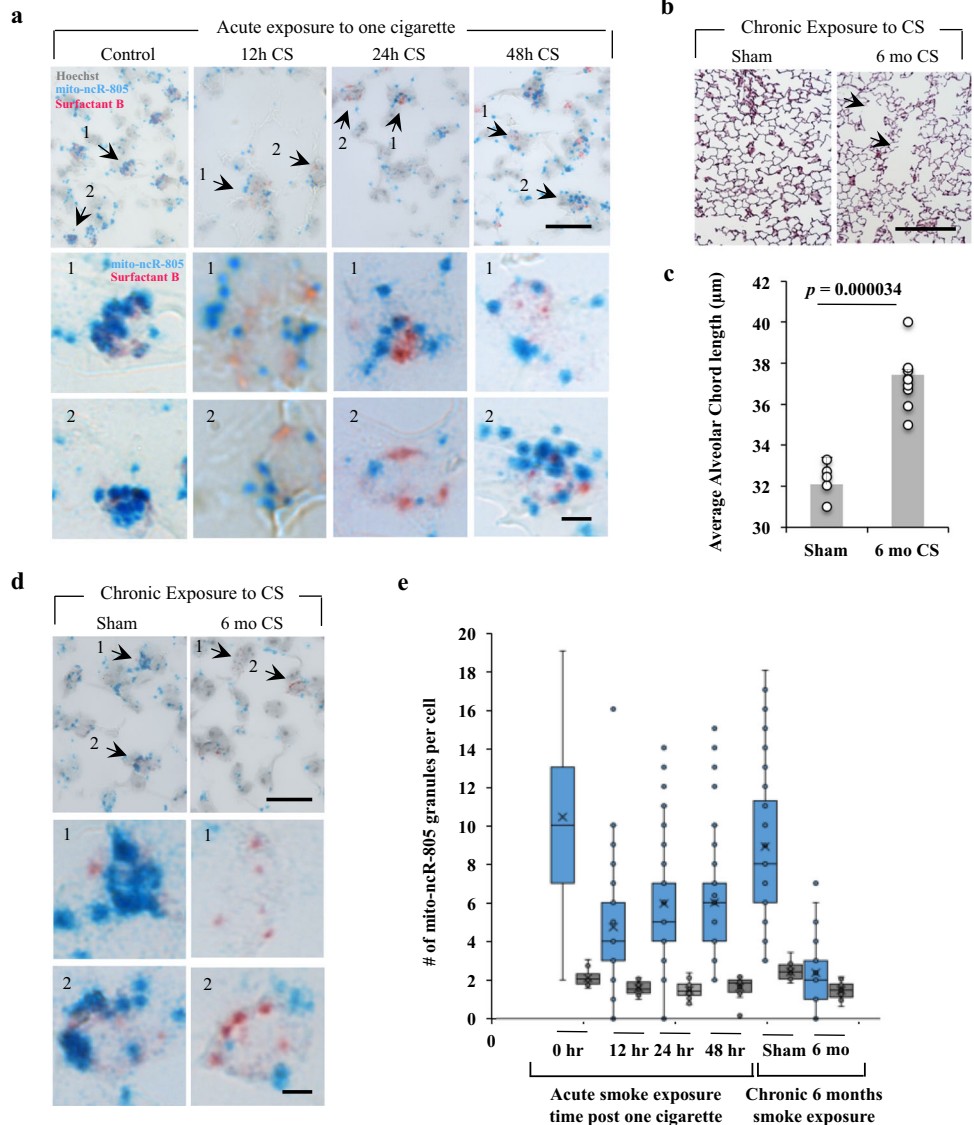

**Fig. 5 The number of mito-ncR-805 spots is decreased in AETII cells of mice acutely and chronically exposed to CS. a–d** Mice were exposed to smoke from one cigarette (**a**) or to smoke from 2 cigarettes per day, 5 days a week, for 6 months (**b–d**). Lungs were inflated, embedded into paraffin blocks, 4 micron sections were cut, and analyzed. **a, d** Expression of mito-ncR-805 (blue) and Surfactant B mRNA (red) using ISH. Images were acquired using ×100 objective. Representative images are shown, scale bar is 20 μm. Arrows indicate enlarged AETII cells shown in (1) and (2) panels, respectively. Scale bar for enlarged images is 5 μm. Hoechst counterstain overlay is removed from enlarged images to maximize contract of mito-ncR-805 and Surfactant B signals. **b** Examples of lungs of sham and mice exposed to CS for 6 months stained with Gill staining to visualize alveolar compartment. Scale bar is 100 μm. **c** Morphometric analysis of average alveolar chord length of sham and 6-month CS-exposed mice, where sham $n = 6$ animals and 6-month CS $n = 9$ animals. **e** Quantification of the number of mito-ncR-805 spots in AETII and non-AETII cells. Each point represents counting from 6 to 8 randomly chosen fields acquired at ×100 magnification from each individual animal. For controls, $n = 76$ AETII and 480 non-AETII cells. For 12 h of CS exposure, $n = 79$ AETII and 403 non-AETII cells. For 24 h, $n = 87$ AETII and 472 non-AETII cells. For 48 h, $n = 53$ AETII and 376 non-AETII cells. For the sham group, $n = 61$ AETII and 322 non-AETII cells. For chronic smokers, $n = 64$ AETII and 414 non-AETII cells. $P$ values are $p\text{Control}^{\text{AETII/non-AETII}} = 5.55558\text{E}{-}08$, $p^{\text{Control/12h-AETII}} = 1.04561\text{E}{-}18$, $p^{\text{Control/12h-non-AETII}} = 0.004329157$, $p^{\text{Control/24h-AETII}} = 1.43455\text{E}{-}13$, $p^{\text{Control/24h-non-AETII}} = 0.002372331$, $p^{\text{Control/48h-AETII}} = 3.89362\text{E}{-}11$, $p^{\text{Control/48h-non-AETII}} = 0.054832469$, $p^{\text{Sham/6 mo-AETII}} = 9.28908\text{E}{-}21$, $p^{\text{Sham/6 mo-non-AETII}} = 0.000104857$, $p6\text{-mo-sm}^{\text{AETI/non-AETII}} = 0.057089623$.

fractionation analysis followed by Northern blot analysis demonstrates disappearance of granular form and intracellular re-distribution of mito-ncR-805, and qPCR demonstrates an increase in the expression levels of mito-ncR-805 due to cellular stress.

**mito-ncR-805 spots are decreased in CS-exposed AETII cells.** Previous research demonstrated that, in the mouse model of COPD, similarly to human smokers, there is an adaptation phase

(lasting up to 4–5 months in the mouse model), which subsequently fails if smoking continues (post-6-month CS exposure), leading to the manifestation of disease phenotypes[48]. We therefore tested whether acute, one cigarette to naive mice (adaptive) (Fig. 5a), or chronic CS exposure associated with emphysema development (Fig. 5b–d) caused any detectable changes in the number of mito-ncR-805 spots. The dynamics of mito-ncR-805 dots in AETII cells in naive mice (Fig. 5a) is similar with that of CSE-exposed MLE12 cells (Fig. 4d); the ISH dots decreased

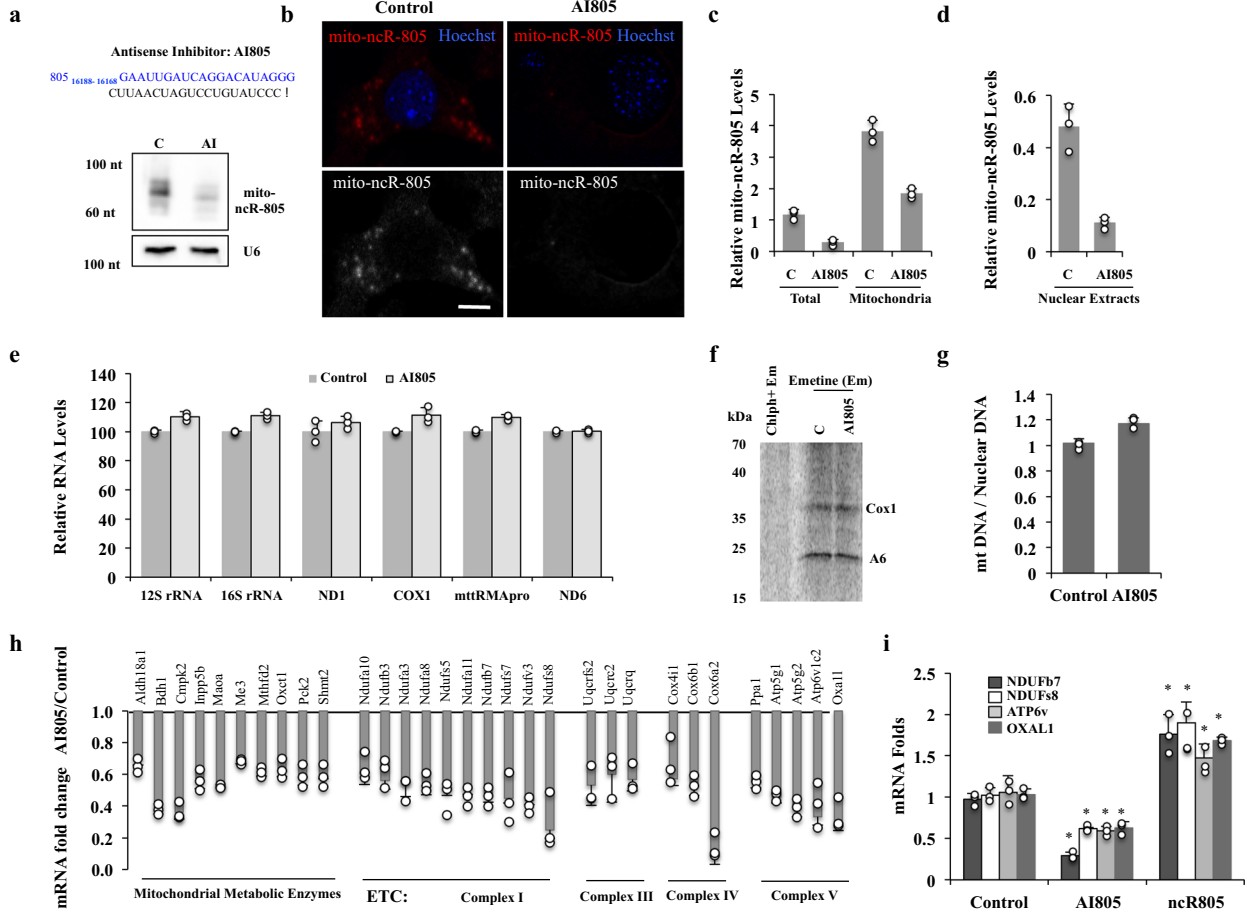

**Fig. 6 Depletion of mito-ncR-805 does not affect steady-state mitochondrial replication, transcription, or translation but a subset of neMITO. a** Antisense inhibitor (AI) of mito-ncR-805 (AI805) was transfected into MLE12 cells to decrease mito-ncR-805 levels. The sequence of the AI805 is shown. Northern blot and **b** FISH analysis of mito-ncR-805. Scale bar 10 μm. **c, d** Cells were transfected with AI805 or non-targeting RNA. Total, mitochondrial, and nuclear extracts were isolated. mito-ncR-805 levels were analyzed by RT-qPCR. **e–g** MLE12 cells were transfected with AI805 or non-targeting RNA. **e** RNA was extracted and the expression of six mitochondrial genes was tested: two encoded by the light strand and four by the heavy strand (see Fig. 2d for their location in mitochondrial genome). **f** Representative phospho-images of mitochondrial translation products labeled with [35S]-methionine and [35S]-cysteine for 60 min following the addition of emetine to inhibit non-mitochondrial translation or the combination of emetine and chloramphenicol to inhibit all translation. **g** DNA was extracted and mitochondrial copy number was determined ($n = 3$ biological experiments for each condition; $p = 0.05$). **h** MLE12 cells were transfected with AI805 or with non-targeting RNA. Levels of mito-ncR-805 in cells transfected with non-targeting RNA were 1.18 +/− 0.2 and in cells transfected with AI805 0.28 +/− 0.04 in 3 independent biological experiments. RNA was analyzed for changes in the expression of neMITO using Qiagen Mitochondrial Energy Metabolism RT$^2$ Profiler PCR Arrays. All gene expression was normalized to GAPDH. Genes with expressed fold changes ≥1.7-fold were grouped based on known function. **i** Cells were transfected with AI805, ncR805, or with non-targeting RNA and analyzed by RT-qPCR for the expression levels of four representative genes found to be downregulated in **h**. Significant $P$ values are marked by asterisks, and $P$ values were calculated from at least 3 representative biological experiments as Control versus AI805 or Control versus ncR805 ($p$Ndufb7$^{C/AI}$ = 7.01036E−05, $p$Ndufb7$^{C/ncR805}$ = 0.004, $p$Ndufs8$^{C/AI}$ = 0.005, $p$Ndufs8$^{C/ncR805}$ = 3.03509E−05, $p$Atp6v$^{C/AI}$ = 0.004, $p$Atp6v$^{C/ncR805}$ = 0.0004, $p$OxaL1$^{C/AI}$ = 0.017, $p$OxaL1$^{C/ncR805}$ = 0.029).

during CS exposure. These data suggest that the release of the mito-ncR-805 from mitochondria is likely occurring in vivo during adaptation to smoking in AETII cells. In non-AETII cells, mito-ncR-805 also decreased in CS-exposed mice, but this decrease was less significant. There was no age-related difference between control mice of acute (8 weeks old) and chronic (8 months old) CS exposure groups, but the number of mito-ncR-805 dots in AETII cells of chronically exposed mice was severely diminished compared to acutely exposed mice (Fig. 5e). In all, 9.2% of AETII cells in chronic smokers showed no mito-ncR-805 dots present (Fig. 5d). The difference between AETII and non-AETII cells became not significant. These data point to a correlation between the number of mito-ncR-805 dots/cell and the well-described diminished ability of AETII cells of chronic smokers to repair CS-induced damage and repopulate lost AETI

cells. The latter is also supported by the development of emphysema in mice exposed to chronic smoking (Fig. 5b, c). Data suggest that the decreased mito-ncR-805 score marks the decline of adaptation to smoking and correlates with disease.

**mito-ncR-805 regulates a subset of nuclear-encoded mito-chondrial genes.** Our ability to manipulate the expression levels of mitochondrial genes is technically limited[31]. Nevertheless, import of RNAs into mitochondria has been reported, although its mechanism is still being elucidated[49]. Moreover, nuclear localization suggests a possibility of a functional role of mito-ncR-805 outside of mitochondria. To initiate studies into the effects of mito-ncR-805, we tested whether we can downregulate its levels using a conventional antisense approach. An antisense inhibitor (AI805) was designed to complement the region of mito-ncR-805

that has no homology elsewhere in the genome (Supplementary Fig. 2g, h). Northern blot and FISH analyses demonstrated that AI805 downregulates the 70-nt mito-ncR-805 transcript (Fig. 6a, b). Since the endogenous mito-ncR-805 localizes to the mitochondria and nucleus, we separated mitochondrial and nuclear extracts to validate the inhibitory action of AI805 in those cellular compartments (Fig. 6c, d). AI805 lowered mito-ncR805 levels in both compartments.

A synthetic full-length mito-ncR-805, ncR805syn, was introduced into the cells. ncR805syn can be detected in the nucleus and causes an increase in mito-ncR-805 dot size (Supplementary Fig. 4). Selected biological effects were tested.

mito-ncR-805 is transcribed from the LSP in the D-loop regulatory region of the mtDNA. To test the possible effect of mito-ncR-805 on mtDNA transcription, the effects of AI805 on the steady-state levels of six mtDNA-encoded RNAs was tested, and no significant effects were found (Figs. 2d–6e). There was also no significant effect on mt-mRNA translation (Fig. 6f). The LSP, in addition to transcription initiation for the expression of light-strand genes, it also generates short RNAs that can prime the replication of the heavy strand of mtDNA[24]. Relative mtDNA copy number was not statistically different between Control and AI805 cells (Fig. 6g). While these assays are not direct measures of transcription and replication rates, the results suggest that mito-ncR-805 likely does not play a major role in mtDNA transcription/replication balance during normal growth. Findings are in agreement with mito-ncR-805 playing a regulatory role in the communication between the mitochondria and the nucleus, since it is leaving the mitochondria and reaching the nucleus during stress (Fig. 4).

Based on our observation that the nuclear accumulation of mito-ncRNA805 decreases with AI805 transfection (Fig. 6d), we asked whether the expression of nuclear-encoded mitochondrial genes (neMITO) is influenced by mito-ncR-805. Expression of these genes was compared in MLE12 cells carrying or not AI805 using pathway-focused gene expression arrays. Scatter plots of 88 genes that function in mitochondrial energy metabolism demonstrate that 14 genes were downregulated (more than twofold change) compared to controls (Fig. 6h and Supplementary Data). Among these genes are multiple subunits of ETC complexes and metabolic enzymes, including enzymes of the TCA cycle and related pathways, such as Maoa, Me3, and Oxct1. In agreement, cells carrying ncR805syn demonstrated increased expression of representative genes, negatively regulated by AI805 (Fig. 6i). It is well established that signaling between mitochondria and the nucleus is needed to maintain mitochondrial homeostasis or to change the composition and function of the organelle[50]. We propose that mito-ncR-805 may act as a retrograde signal from mitochondria to the nucleus to achieve this.

**mito-ncR-805 modulates mitochondrial metabolism**. To establish the effect of mito-ncR-805 inhibition and enhancement on the activity of the TCA cycle, we directly measured metabolic flux using stable isotope incorporation and mass isotopomer analyses. AI805 had no effect on the rate of $^{13}$C-label incorporation into lactate and pyruvate, a measure of glycolytic rate (Fig. 7a and Supplementary Data). However, we observed a substantial decrease in the incorporation of $^{13}$C-label from [U-$^{13}$C]glucose into all tested intermediates of the TCA, including the acetyl-CoA and OAA moieties of citrate, alphaKG, succinate, fumarate, and malate (Fig. 7b). The threefold decrease in carbon flux into the committed steps of TCA, indicated by the decreases in the formation of bona fide TCA substrates aspartate and glutamate, is suggestive of decreased cellular respiration that in turns leads to carbon preservation and to decreased cataplerosis (Fig. 7c).

Conversely, ncR805syn increased glycolytic rate, as shown by the doubling in pyruvate and its by-products, lactate and alanine (Fig. 7a, c). ncR805syn increased carbon flux through the TCA cycle (Fig. 7b). Cells carrying ncR805syn had significantly increased flux of succinate, indicating strong forward TCA cycle activity. An increase in TCA cycle flux would require extra oxaloacetate to maintain citrate formation; indeed, significantly higher OAA flux was observed in cells carrying ncR805syn. Therefore, ncR805syn promotes a significant metabolic response, and results presented above support the reciprocal effects of inhibition and stimulation on TCA cycle activity by mito-ncR-805 (Fig. 7d). Although MLE12 cells are glycolytic at baseline (average 40 µmols/day of glycolytic, versus 0.4 µmols/day of mitochondrial rate), the ability to generate extra ATP through mitochondrial respiration may be critical for survival in response to stress.

**mito-ncR-805 modulates mitochondrial bioenergetics**. We measured the mitochondrial oxygen consumption rate (OCR) at baseline and in response to mito-ncR-805 inhibition by AI805 using XFp Cell Mito Stress Test (Fig. 7e). AI805 had inhibitory effects on major mitochondrial respiratory functions: maximal mitochondrial respiration, spare respiratory capacity, coupling efficiency, and ATP production (Fig. 7e, and Supplementary Data —calculations). AI805 also lowered basal respiration and increased proton leakage, making mitochondrial respiration less efficient. The rates of non-mitochondrial respiration were not affected, consistent with the metabolomics data (Fig. 7a, d).

Low activity of the TCA cycle, and ETC, should result in lower membrane potential ($\Delta\Psi$m)[51]. We measured $\Delta\Psi$m[52] and found that AI805 treatment correlates with lower $\Delta\Psi$m (higher depolarization) (Fig. 7f). Together, these data indicate that mitochondrial respiration and membrane polarization are affected by lowering mito-ncR-805. Cells with a larger potential pool of ATP (high maximal and spare respiratory capacities, coupling efficiency, and $\Delta\Psi$m) should be able to channel more energy to repair damages and replace lost cells. We examined how control and cells carrying AI805 repopulate lost cells after the scratch-wound assay. Cells carrying AI805 demonstrated slower wound healing than control (Fig. 7g, h). The data are in agreement with the idea that mito-ncR-805 stimulates cellular pathways involved in energy metabolism and cell growth.

## Discussion

We have identified an ncRNA, mito-ncRNA-805, that is encoded by mtDNA. It is upregulated in response to CS exposure during the adaptive phase in AETII cells in a mouse model of COPD and in primary AETII cells in culture. During the stress of smoking, mito-ncR-805 demonstrates spatial and temporal dynamics that can be divided into three major steps: redistribution from its mitochondrial localization, increase in expression levels, and appearance in the nucleus. The positive effects of mito-ncR-805 on mitochondrial homeostasis and function delineated herein suggest that mito-ncR-805 induction is a defense mechanism in response to stress in AETII cells. We propose that mito-ncR-805 translocates from mitochondria to the nucleus to induce a nuclear response that promotes mitochondria bioenergetics (Fig. 7i). This bioenergetics helps cells to survive the stress.

The existence of regulatory small ncRNAs encoded by the mtDNA has been reported[26–28]. Their functions are being elucidated. The involvement of some in communication between mitochondria and the nucleus was suggested. For example, the mito-lncRNA, SncmtRNA, is observed in both the mitochondria and the nucleus and is shown to function in retrograde signaling[30]. How those ncRNAs are imported outside of mitochondria and exported into the nucleus is not known. A few vague possible

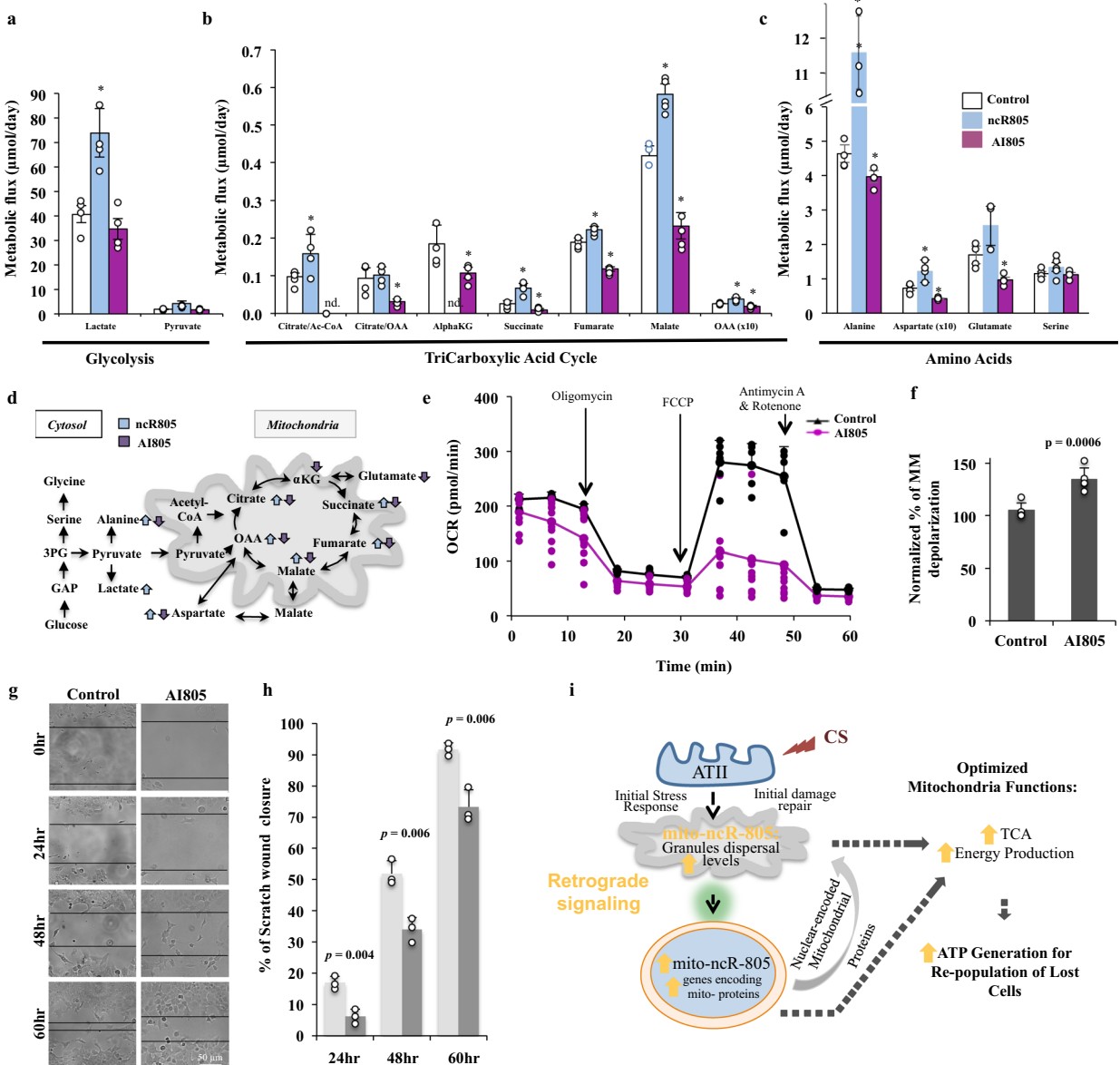

**Fig. 7 mito-ncR-805 modulates mitochondrial metabolism and bioenergetics. a–d** MLE12 cells were transfected with AI805, ncR805, or non-targeting sequence. At 30 h post-transfections, cells were exposed to 20 mM [U-$^{13}$C]glucose for 12 h, harvested, and metabolic flux was determined following $^{13}$C label incorporation into various metabolites. **a** Metabolic flux of glycolysis, **b** TCA, and **c** amino acid turnover. Metabolic flux was determined as a product of fractional flux multiplied by the concentration (pool size) of a metabolite. Fractional flux was calculated as $^{13}$C enrichment of a product metabolite/[$^{13}$C] enrichment of precursor, i.e., [U-$^{13}$C]glucose. All quantifications are presented as mean of four independent experiments ± S.E.M (Supplementary Data). Graphs are representative of three (for AI805) and two (ncR805) independent biological experiments. Asterisks denote statistically significant changes. **d** Schematic representation of the major findings in **a**, **b**. **e** MLE12 cells were transfected with AI805 or non-targeting RNA at 48 h post-transfections; OCR were measured at basal level with subsequent and sequential addition of oligomycin, FCCP, and rotenone + antimycin A, using a Seahorse XF$^e$96 extracellular flux analyzer. Graph is a representative of one out of three biological experiments. All three experiments have been used for the quantification of bioenergetics parameters (Supplementary Data). **f** Mitochondrial membrane potential was measured using the JC-10 assay, $n = 5$ biological experiments, $p$ value = 0.0005. **g** Scratch injury was inflicted to MLE-12 cells transfected with AI805 or non-targeting sequence and grown to the confluence of 80%. Live Images were acquired at 0, 24, 48, and 60 h post-scratch using a Nikon Eclipse TE6200-LI microscope. **g** is a representative of three independent experiments. **h** Quantification of healing speed of **g**. **i** Schematic representation of the findings reported in this manuscript. Following exposure of AETII cells to CS, the mtDNA-encoded mito-ncR-805 is released from mitochondria; its levels increase in mitochondria and in the nucleus. High levels of mito-ncR-805 in the nucleus correlate with increased expression of nuclear genes, which encode mitochondrial proteins related to mitochondrial homeostasis and function. Increased expression of those proteins ensures high bioenergetics of the mitochondria, which supports faster repopulation of lost due to damage cells.

mechanisms have been suggested, such as an existence of yet unidentified transporter, piggyback mechanism, and re-purposing of Tom-Tim translocases[27,31,49]. The way mito-ncR-805 traffics within the cell is not clear. The increased co-localization with

Tom20 during stress may be a hint toward the re-purposed translocase hypothesis.

mito-ncR-805 may represent an unclassified subtype of mito-ncRNAs that is induced in response to cellular stress. In fact, to

our knowledge, the discovery of mito-ncR-805 provides the first evidence of the function of the control region of the mitochondrial genome beyond mitochondria genome regulation.

mito-ncR-805 is immediately adjacent to and transcribed from the LSP and the D-loop regulatory region of the mtDNA. Two main types of transcripts are generated from the LSP start site: polycistronic full-length light-strand transcript and shorter transcripts terminated in the area of CSBII, and used to prime replication of the heavy strand of mtDNA[18,19,42]. mito-ncR-805 can be processed from the full-length polycistronic transcript, or alternatively, it can be processed from the terminated CSBII transcripts, and stored for future use. There is therefore an undiscovered mechanism to accumulate mito-ncR-805 for use in stressed cells.

We found that mito-ncR-805 accumulates in mitochondrial granules. Mitochondria granules have been previously described as centers for posttranscriptional mRNA processing[53,54]. It is possible that mito-ncR-805 is stored in some type of mitochondrial granules, which may serve as "a stress-response hub" of ncRNAs generated by the mitochondrial genome. mito-ncR-805 may be the first reported member of this hub with the rest awaiting their discovery. Our findings point to the existence of a mitochondria-initiated mechanism, which shapes the cellular response to stress. This idea is supported by findings that the mitochondrial protein DELE1 is activated during mitochondrial stress, relocates to the cytosol, and communicates mitochondrial stress into a cytoplasmic response, the phosphorylation of eIF2α[55,56].

Expression of mito-ncRNAs has been demonstrated to have cell-type-specific patterns, and their regulation and possibly function have been suggested to be cell-type specific[26,28]. Our data support this idea. We found that the expression of mito-ncR-805 is higher in AETII cells than in non-AETII cells. Mito-ncR-805 redistributes from its mitochondrial localization during stress, concomitantly with an increase in its total levels in AETII cells. The expression levels of mito-ncR-805 decrease with CSE-induced injury in total lung tissue and in alveolar macrophages, cells which have high turnover rate in COPD and are mainly lost during CS exposure[3]. Analogous to our total lung analysis, others have reported that miR-805 (possibly mito-ncR-805) is down-regulated in the necrotic renal ischemic reperfusion injury model in total kidney[57]. We detected upregulation of mito-ncR-805 in local lung progenitor cells[19] and livers, where high levels of mito-ncR-805 may ensure the adaptive response and better and longer periods of survival of those cells during multiple stressful challenges. This correlates with findings on the role of these cell types in COPD and implies that preservation of mitochondrial function by upregulation of mito-ncR-805 may be a general mechanism in progenitor cells. Its exhaustion from mitochondrial granules marks the disease, which correlates with advanced age. It would be interesting to determine whether local kidney progenitor cells differ in their mito-ncR-805 expression in response to ischemic reperfusion injury, similar to the difference between local lung progenitors from total lung. mito-ncRNAs that originate from the 16,188–16,162 region of the mtDNA were found to be upregulated in hippocampal olfactory neurons during discrimination learning in mice[58]. It has been suggested that this may link behavioral plasticity to the increase in protein synthesis required during the learning process, which requires energy. These examples point to the cell-type-specific regulation of expression and possibly processing and broad physiological function of transcripts containing the mito-ncR-805.

Regulation of mito-ncR-805 during chronic CS stress is a complex multi-step process that involves upregulation of its levels, redistribution from its mitochondrial localization, and nuclear localization. The upstream mechanisms that regulate these steps are unknown. CS is a complex mixture of nicotine and multiple free radicals. Several components of CSE may influence different steps in mito-ncR-805 regulation. Nicotine has pro-survival effects, with cell-type-specific responses that depend on the repertoire of nicotinic acetylcholine receptors (nAChRs) on the cell surface and on the mitochondrial membrane itself in the given cell type[59–61]. We do not know whether a single component of CSE is responsible for all spatial–temporal changes in mito-ncR-805, or the response is a complex reaction that culminates from the effects of several CS components, or is a result of a signaling event through nAChRs or other mechanisms.

Why do we still observe cell death in lung parenchyma of COPD patients? The cell loss means that existing AETII cells lose the ability to replenish dying AETI cells. Indeed, the expression score of mito-ncR-805 in AETII cells from chronic CS-treated mice that developed emphysema drops significantly, with no mito-ncR-805 signal detected in 9% of the cells. A possible explanation is advanced age of COPD manifestation. Aging is characterized by decreased cellular adaptation; mito-ncR-805 upregulation may be lost in aged cells. Future research will clarify whether decline in mito-ncR-805 can serve as an indicator of failure of mitochondrial function in COPD.

The mechanism of nuclear action of mito-ncR-805 is not known. Regulation of gene transcription and/or mRNA turnover are possible mechanisms. Some aspects of the nuclear function of mito-ncR-805 on energy production may be via NRF1, major factor in nuclear–mitochondria communication. NRF1 is a known inducer of neMITO[50]. Based on known targets of NRF1 identified by chromatin immunoprecipitation–sequencing studies[62], and the limited number of genes we identified to be regulated by mito-ncR-805, it is tempting to speculate that an NRF1-related mechanism may be operating in our model.

Similar signaling events from mitochondria to the nucleus are mediated by short mitochondria-derived peptide MOTS[63,64]. MOTS senses mitochondrial status and signals it to the nucleus. Similar to dynamics of mito-ncR-805 during CS-induced stress, basal levels of MOTS exist in the nucleus but increase significantly during oxidative stress. In the nucleus, MOTS activates transcription of an antioxidant defense factor NRF2. MOTS retrograde signaling differs from that of mito-ncR-805 in many aspects: it is mediated by a peptide rather than an ncRNA; it occurs earlier during oxidative stress and aims to restore mitochondrial homeostasis, while mito-ncR-805 signals that mitochondrial homeostasis is restored, and functions can resume. Significant amounts of MOTS can be found in the cytosol, where its mRNA is translated. Very small, if any, amounts of mito-ncR-805 are cytosolic. Despite the differences in their respective signaling, the discoveries of MOTS and mito-ncR-805 provide strong evidence for the idea that signaling occurs between mitochondria and nucleus, which relays the functional state of mitochondria and activates mechanisms outside of the mitochondria accordingly.

The findings here of an mtDNA-generated RNA-mediated regulation of mitochondrial health and integrity open a new dimension in our understanding of the regulation of mitochondria function in health, with implications in multiple diseases characterized by mitochondrial malfunction.

## Methods

**MLE12 cell line**. MLE12[33] cell line were purchased from and cultured per American Type Culture Collection (ATCC) instructions. Briefly, MLE12 cells were cultured in RPMI complete media: RPMI1640 medium (Cellgro, Manassas, VA, USA) supplemented with 4% fetal bovine serum (FBS; Serum International, Laval, QC, Canada), insulin–transferrin–selenous acid (ITS) premix (BD Biosciences, San Jose, CA, USA), 10 nM hydrocortisone (Sigma-Aldrich), 10 nM estradiol (Sigma-Aldrich), 10 mM HEPES, 2 mM glutamine (CellGro), 100 U/ml penicillin, and

100 g/ml streptomycin. All experiments were performed on cells between passages 2 and 12.

**Mice.** C57BL/6 female mice (8–10-week old) from Taconic Farms (Germantown, NY, USA) were used for experiments. All mice were housed and used for the experiments under the direction and approved protocols of the Mercer University Institutional Animal Care and Use Committee (protocol A1009016).

**Acute and chronic CS exposure in vivo.** Mice were exposed to one unfiltered cigarette for acute or to two unfiltered cigarette, twice a day, for 5 days a week for the indicated periods of times for chronic CS exposure or to a sham, using the smoking chamber for the nose only exposure, also some minute whole-body exposure cannot be absolutely ruled out, as described previously[65,66]. For both groups, exposure started at the age of 8–10 weeks.

**Tissue processing and collection.** Tissue processing and collection was done as described[36,65]. Briefly, animals were terminally anesthetized by a combination of ketamine/xylazine (100/5–10 mg/kg) followed by cervical dislocation. Lungs and livers were removed, flash-frozen, and stored at −80 °C for further analysis. For chronic smoking protocols, lungs were harvested 16 h post last CS exposure.

**Tissue processing for ISH analysis.** Lungs were inflated by instilling intra-tracheally a 10% neutral buffered formalin at 25 cm $H_2O$ for 10 min. They were then ligated and removed. Inflated lungs were fixed in the formalin solution for 24 h, followed by 48 h of cold phosphate-buffered saline (PBS) wash and paraffin embedded. Serial sagittal sections of 4 μm were obtained for analysis.

**Cell viability.** Cell viability was tested using an MTT-based in vitro toxicology assay or in vitro toxicology assay kit neutral red based (Sigma Aldrich, TOX1-1KT and TOX-4) or CellTiter-Glo 2.0 cell viability assay kit (Promega), according to the manufacturer's instructions.

**Primary AETII cell isolation and culturing.** Primary AETII cell isolation and culturing were done as previously described with some modifications[67].

*Isolation.* Mice were anesthetized with an intraperitoneal injection of ketamine (100 mg/kg) and xylazine (8.5 mg/kg). The abdomen was opened by midline incision and mouse was exsanguinated by cutting the descending aorta. The lungs were exposed and perfused by injecting PBS into the left ventricle, 1 ml of Dispase (50 U/ml BD Biosciences, cat # 354235) was injected into the lung with a 1-ml syringe connected to the cannula placed in the trachea, ligated, removed, placed in 3 ml Dispase in a 15-ml sterile conical centrifuge tube, and incubated at room temperature (RT) for 25 min. The lungs were then moderately homogenized in a 60-mm Petri dish with 3 ml Mouse Wash Media (MWM: Dulbecco's modified Eagle's medium (DMEM)/F12, with addition of MEM non-essential amino acids and Pen-Strep) with 5 mg/50 ml DNase (MWM/DNAse) using forceps. The cell suspension was filtered through 100-, 70-, 40-, and 20-μm filters (BD Biosciences), rinsing each filter with 2 ml MWM/DNAse. Cell were pelleted by centrifugation at 1200 rpm for 5 min at 10 °C, suspended in 10 ml MWM/10% FBS, and plated into 100 mm cell culture dishes for 30 min to allow for fibroblasts to adhere to the surface. Supernatant was collected, and fibroblast selection was repeated. The cell suspension was further enriched for Type-II cells by negative selection against unwanted cell types by incubation for 30 min at 37 °C with biotinylated antibodies against the particular cell type: α-CD-16/32 (BD Biosciences, 10 μl of 0.5 mg/ml) for macrophages, α-Ter-119 (BD Biosciences, 15 μl of 0.5 mg/ml) for erythroid cells; and α-CD45 (BD Biosciences, 20 μl of 0.5 mg/ml) for leukocytes. Streptavidin-coated magnetic beads (Promega, Madison, WI) were added to the mixture to remove the complexes of cell–antibody–Streptavidin-coated magnetic particles. Remaining cell suspension was plated on IgG (Innovative Research, Novi, MI) covered plates for 2 h at 37 °C to allow macrophages to adhere. Non-adherent cells were collected, pelleted, and re-suspended in 1 ml of HITES medium (Ham's F12 with 15 mM HEPES, 0.8 mM $CaCl_2$, ITS, 10 nM hydrocortisone, and 10 nM of 17-beta-estradiol). Cells were either plated at the concentration of $5 \times 10^4$ per well in a 1:5 Matrigel (BD Biosciences)/HITES medium coated 24-well plate or spun onto slides using a Cytospin (Thermo Scientific), at the density of $1.5 \times 10^5$ cells per slide. Immunofluorescent staining for laminar body marker, LC3, in conjunction with nuclear staining with Hoechst was used to see efficiency of purification.

*Culturing.* Cells were cultured in a MWM containing DMEM/F12 (Life Technology) and 100 U/ml penicillin and 100 g/ml streptomycin, with or without 0.01% DNAse (Sigma Aldrich); an MWM containing 10 FBS; and a HITES media containing Ham's F12 (Life Technologies), 10% FBS, 15 mM HEPES, 0.8 mM$CaCl_2$, ITS, 10 nM hydrocortisone, and 10 nM 17-beta-estradiol.

*Immunofluorescent staining.* Cells grown on 12 mm coverslips were fixed using 3.8% paraformaldehyde (Sigma Aldrich), permeabilized by PBS supplemented with 0.05% Saponin (Sigma Aldrich), blocked using 5% goat serum, incubated with

specified primary antibody, diluted as indicated for each antibody, in PBS supplemented with 0.05% Saponin for 2 h at RT, washed with PBS supplemented with 0.05% Saponin three times, and incubated with an appropriate secondary antibody. Hoechst (Sigma-Aldrich) staining was used to visualize nuclei. The primary antibodies used to detect Type-II cells were anti-ATP-binding cassette subfamily A member 3 antibody [3C9] (Abcam, Cambridge, UK, 1:100) and secondary antibody goat anti-Mouse IgG Alexa Fluor 594 594 (Molecular Probes, Cat. # A11032, 1:500). Cells were mounted using Gelvatol mounting media (https://doi.org/10.1101/pdb.rec10252 Cold Spring Harb Protoc 2006).

**Transmission electron microscopy (TEM).** Isolated AETII cells were plated, allowed to adhere and recover from isolation procedure for 3 days, fixed using Karnovsky's fixation protocol, and processed for TEM to visualize laminar bodies. MLE12 cells exposed or not to CSE were processed as described above to visualized mitochondria. Images were acquired using JEM 1011 TEM microscope. Final images were processed using Adobe Photoshop CS5.1 (Adobe Systems Incorporated).

**miRNA array data analysis.** Raw miRNA microarray data was analyzed using GeneSpring GX 11.0 (Agilent Technologies). Data was $\log_2$ transformed, signal values were normalized and baseline transformation to the control samples was performed. Data were classified into specific groups within the program. Statistical analysis of the data for differential expression and significances of observations was performed using $T$ test.

**Preparation of CSE.** Preparation of CSE was described by Kenche et al.[36] Research-grade 1R5F cigarettes (Kentucky Tobacco and Health Research Institute, Lexington, KY, USA) were used to prepare the CSE. CSE was made fresh for each experiment by bubbling the smoke from 1 cigarette through 5 ml of serum-free medium in a 15-ml conical tube and filtering it through a 0.22-μm filter to remove large particles and maintain sterility. This solution was designated as 100% CSE and diluted in culture medium to yield concentrations specified for each experiment.

**miRNA and mRNA extraction from cells and tissues.** miRNA and mRNA extraction from cells and tissues was done using the miRNeasy and mRNeasy Kits (Qiagen). Cells were washed twice with PBS and lysed using Qiazol reagent (Qiagen, Valencia, CA, USA). Tissues, size of a couple of rice grains, were homogenized in 1 ml of Qiazol. miRNA was extracted using the miRNeasy Kit (Qiagen) and mRNA was isolated using the mRNeasy Kit (Qiagen) as per the manufacturer's protocol. cDNA synthesis for both miRNA and mRNA was carried out using the High Capacity cDNA Reverse Transcription Kit (Applied Biosystems, Foster City, CA, USA).

**Quantitative real-time (qRT) polymerase chain reaction (PCR).** cDNAs were amplified by qRT-PCR using Applied Biosystems 7500. Primer/probe Taqman assays were purchased from Life Technologies Thermo Fisher Scientific for the quantitation of miRNA: mmu-miR-805 (002045), mmu-miR-709 (001644), mmu-miR-1195 (002839), mmu-miR-1907 (12ii42), and custom mito-ncR-805[21–57 bp] (CSXOZ7P) and human miR-805/mitosRNA L DL-1 (CS39QW4). Taqman Assay sno55 (001228) and RNU48 (001006) were used as a housekeeping control for quantitation in mouse and human, respectively. All Taqman assay qRT-PCRs were performed using TaqMan Universal Master Mix II, no UNG from Life Technologies (4440040). All original data sources are provided in Supplementary Data Sheet 2.

**Separation of mitochondrial and cytosolic fractions.** Separation of mitochondrial and cytosolic fractions was done using the Mitochondrial Isolation Kit from Sigma (MITOISO2, Sigma, Saint Louis, MO, USA) according to the manufacturer's instructions for isolation of mitochondria from cells with modifications. Briefly, cells grown on plates were washed twice with PBS, scrapped into lysis buffer, which is 1× Extraction buffer supplemented with protease inhibitory cocktail (Sigma-Aldrich Roche Biochemical Reagents) and 1:200 dilution of cell lysis solution, incubated 5 min on ice, and diluted with two volumes of 1× Extraction buffer. Lysates were centrifuged at $600 \times g$, 10 min, at 4 °C. Third of the supernatant was removed for analysis of Total fraction, which was further divided into two, one for protein analysis, and the second one for RNA extraction. The remaining of the lysates were centrifuged at $11,000 \times g$ for 10 min, at 4 °C. Supernatant was carefully removed and saved for further analysis as cytosolic fraction, which was divided further into two for protein and RNA analysis. Pellets were treated with RNase A/ T1 mix (0.5 U/20 U/100 μl), at 37 °C, for 30 min to eliminate cytosolic RNAs, washed, and suspended in 1×Extraction buffer; half was used for western blot analysis, the other half for RNA extraction and analysis.

**Protein lysate preparation and western blot analysis.** Protein lysate preparation and western blot analysis were done as described by Kenche et al.[36] The following primary antibodies were used: α-Ago2 (Wako Chemicals, Cat. # 018-22021, 1:500), α-tubulin (Millipore-Sigma, Cat. # T9026, 1:5000), α-lactate dehydrogenase A, (Cell

Signaling, Cat # 2012, 1:1000), α-succinate dehydrogenase subunit A, (Abcam, Cat # ab14715, 1:1000), α-H3 (Cell Signaling, Cat # 9715, 1:4000), α-GADD34 (Santa Cruz, Cat. # SC-825, 1:500), α-Actin (Abcam, Cat. # ab6276, 1:10,000), α-Phospho-eIF2α (Ser51) (Cell Signaling, Cat. # 3597, 1:1000), α-eIF2α total (Cell Signaling, Cat. # 9722, 1:1000), α-Trib3 (LSBio, LifeSpan BioSciences, Cat. # LS-C164592, 1:250), α-Phospho-Akt (Trh308) (Cell Signaling, Cat. # 2965, 1:500), and α-Akt total (Cell Signaling, Cat. # 9272, 1:1000). Secondary antibodies were goat anti-mouse IgG Horse Radish Peroxidase (Thermo Fisher Scientific, Cat # 31430, 1:5000), and goat anti-rabbit IgG Horse Radish Peroxidase (Thermo Fisher Scientific, Cat # 31460, 1:5000). All original blots are provided in Supplementary Fig. 5.

**Northern blot procedure**. Northern blot procedure was done using the Signosis High Sensitive miRNA Northern Blot Assay Kit (NB-1001 and 1002, Signosis, Inc. Santa Cruse, CA, USA). Probes were 5′-biotin-CCCTATGTCCTGATCAATTC for detection of mito-ncR-805 and 5′-biotin-ATCGTTCCAATTTTAGTATATGT GCTGCCGAAGCGAGCAC-3′ for U6 (cat.# MP-0512, Signosis). Northern blots were probed with mito-ncR-805, stripped by incubating in 0.5% sodium dodecyl sulfate, at 60 °C for 60 min, and re-probed with U6 probe for a loading control. All original blots are provided in Supplementary Fig. 5.

**shRNA-Ago2 and shRNA-DICER depletion**. shRNA-Ago2 and shRNA-DICER depletion was done as described previously[68]. Briefly, shControl was from Millipore-Sigma (pLKO.1, SHC001). shAGO2 and shDICER were from Millipore-Sigma (TRCN#255786 and TRCN#71321, respectively) Sequence for Ago2 is CCGGATCGAACATGAGACGTCTTTGCTCGAGCAAAGACGTCTCA TGTTCGATTTTTTG and for DICER CCGGGCTCGGGATGATGGTAAGA GAACTCGAGTTCTCTTACCATCATCCCAGCTTTTTG. Ago2 primers are as follows: amplicon size 138, Forward Primer TGAAGAACACATACGCTGGC, Reverse Primer TCTGCACGTTCTTCATCTGG. DICER TaqMan assay Mm052172_m1. Ago2 antibody was from Wako Chemicals USA (Cat. 018-22021); α-tubulin antibody was from Millipore-Sigma (Cat. T9026). For shRNA knock-down experiments, lentiviral particles expressing shRNA against control and Ago2 mRNAs were prepared and propagated in HEK293T cells. After two rounds of lentiviral infection, cells were selected under puromycin (2 µg/ml) for 2 days. Selected cells were separated to plates with media in the absence of puromycin and followed by designated treatment.

**Mitochondria depletion**. Mitochondria depletion was done as previously described[46]. Briefly, MLE12 cells were grown in the complete RPMI medium supplemented with ethidium bromide (400 ng/ml), pyruvate (1 mM), and uridine (50 µg/ml) for two to six passages.

**Mitochondrial DNA copy number**. Mitochondrial DNA copy number was evaluated as a ratio of mitochondrial to nuclear DNA by qPCR[69]. Ratio of untreated culture was considered as 1. Primers for nuclear DNA were for beta-2-globin genomic locus, for mtDNA were for 16S rRNA gene (assays ID Mm00047666_cn, and ID Mm04260181_s1, Thermo Fisher Scientific).

**Small RNA sequencing**. Small RNA libraries were made according to Heyer et al.[70]. Briefly, RNA was isolated from the two biological replicates of control and two of 10% CSE-treated MLE cells using Trizol (15596018, Ambion). The samples were treated with DNase I (Sigma, 04716728001), re-isolated using Trizol, and RNA quality was checked using Agilent Bioanalyzer. The RNA was further treated with CIP (neb, M0290S), labeled with γ32P[ATP] using T4 PNK (neb, M0201S) and resolved on the 10% TBE-Urea polyacrylamide gels. Fifteen–100-nt RNAs were isolated and used for the construction of libraries. Twenty-two-nt 3′-DNA adaptor (/5Phos/NNNNCTGTAGGCACCATCAAT/3ddC) was ligated to the isolated RNAs using RNA-ligase T4 RNA-Ligase 2 (NEB, M0373S, RNL2 truncated KQ). Each small RNA sample was reverse transcribed with a different, 102-nt-long DNA primer (from RT-1 to RT-4). Each RT primer contained unique 3-nt sample barcode (in red) to identify it after pooling and sequencing:

RT-1: (Phos)
NNNCTAAGATCGGAAGAGCGTCGTGTAGGGAAAGAGTGTAGATCTCG GTGGTCGC(SpC18)CACTCA(SpC18)TTCAGACGTGTGCTCTTCCGATCTA TTGATGGTGCCTACAG,

RT-2: (Phos)
NNNAGCAGATCGGAAGAGCGTCGTGTAGGGAAAGAGTGTAGATCTC GGTGGTCGC(SpC18)CACTCA(SpC18) TTCAGACGTGTGCTCTTCCGATCTATTGATGGTGCCTACAG,

RT-3: (Phos)
NNNATTAGATCGGAAGAGCGTCGTGTAGGGAAAGAGTGTAGATCTC GGTGGTCGC(SpC18)CACTCA(SpC18) TTCAGACGTGTGCTCTTCCGATCTATTGATGGTGCCTACAG, and

RT-4:(Phos)
NNNCCGAGATCGGAAGAGCGTCGTGTAGGGAAAGAGTGTAGATC TCGGTGGTCGC(SpC18)CACTCA(SpC18) TTCAGACGTGTGCTCTTCCGATCTATTGATGGTGCCTACAG.

cDNA was circularized using CircLigase ssDNA-ligase (CL411K, Lucigen, USA). PCR was done with HotStart ReadyMix from KAPA Biosystems (KK2602) using the following primers: Seq F: AATGATACGGCGACCACCGAGATCTACAC, and Seq R: CAAGCAGAAGACGGCATACGAGATGTGACTGGAGTTCAGACGTGT GCTCTTCCG. Quality of the libraries was checked using Agilent HS DNA Bioanalyzer. Paired-end sequencing was performed on an Illumina NextSeq platform (PE 2 × 75 bp). FASTQ files were trimmed of library adapter sequences and converted to FASTA files.

*To determine mito-ncR-805 sequence*. To determine mito-ncR-805 sequence, BlastN analysis was then performed against the target mouse mitochondrial sequence (mtDNA 16,116–16,210). The single best BLAST hit for each read was used in the summary data (duplicates were removed, and any remaining hits were filtered to the single hit with the highest identity). After reverse complementing any BLAST hits on the opposite strand (to ensure the same orientation), the lengths of the sequence alignments were tabulated and counted.

*To determine nuclear homology of mito-ncR-805*. To determine nuclear homology of mito-ncR-805, all the sequencing reads were aligned against the entire mm10 mouse genome using BLASTN, utilizing an *e*-value of 0.00001.

**RNA in situ hybridization (ISH), FISH**
*Probe design*. All FISH and ISH probes were designed by ACD via the ACD probe design software and offers simultaneous signal amplification and background noise suppression, resulting in a very specific signal. To detect mito-ncR-805, a 1ZZ probe (BA-Mm-mt-D-loop-O1-1zz) targeting 1–57 to 16,119–16,188 of NC_005089.1 was generated. To detect Surfactant B mRNA, a 3ZZ probe (BA-Mm-Sftpb-3EJ) targeting 50–582 of NM_147779.2 was generated. For both probes, cross-detection threshold is >95%.

*RNA FISH*. RNA FISH was performed using the BaseScope Detection Red v2 assay (ACD, Inc.) following the manufacturer's protocol. Cells were washed with PBS and fixed in 3.8% PFA for 30 min at RT. Fixed cells were permeabilized and dehydrated in series of ethanol concentrations (50% EtOH–PBS, 70% EtOH, 100% EtOH), for 1 min at RT in each, and stored at −20 °C till further use. Cells were treated with 1:15 dilution of Protease III provided with the kit for 7 min at RT. Cells were then hybridized with BA-Mm-mt-D-loop-O1-1zz-st probe, and amplification steps were performed following the manufacturer's instructions, except that AMP 7 step was 10 min. After hybridization and amplification, cells were incubated with Hoechst solution (1 mg/ml in water, Sigma-Aldrich) to counterstain nuclei and mounted on a microscope slide. Introduction of FastRed pigment into the amplification tree as a visual identifier of the probe binding allows both fluorescent and chromogenic detection. Leica DM IRE 2 microscope operated by the AF600/LAS AF 3.100 software was used to acquire fluorescent non-confocal images. For fluorescent detection, FastRED signal was excited at 560 nm with fluorescence emission collected from 570 to 630 nm. Leica SP8-HyVolution laser scanning confocal microscope (see below) was used to acquire confocal images as specified for each experiment. All images were acquired using identical microscope settings. Images were further processed using the Adobe Photoshop CS5.1 software (Adobe Systems Incorporated).

*RNA FISH-IF*. RNA FISH-IF (immune-fluorescent protein labeling) was performed using the modified BaseScope Detection Red Assay (ACD, Inc.), followed by regular IF detection of Tom20. FISH was done as described above, except Protease III treatment and amplification step AMP7 were shortened to 5 min. Fish was followed by regular IF protocol. Briefly, slides were washed with 0.05% Saponin–PBS (PBSS), incubated with 5% goat sera in PBSS for 1 h to block non-specific labeling, followed by incubation in PBSS supplemented with antibodies specific for Tom20 (Cell Signaling 42406, dilution 1:40) for 2 h, washed 3× with PBSS, incubated with anti-rabbit conjugated to Alexa Fluor 488 (Molecular Probes, dilution 1:100) for 1 h, and washed 3× with PBSS and 1× with PBS. Nuclei were visualized with Hoechst (Sigma-Aldrich). Cells were mounted using Gelvatol mounting media (https://doi.org/10.1101/pdb.rec10252 Cold Spring Harb Protoc 2006).

*Confocal microscopy*. Imaging was performed using a Leica SP8-HyVolution laser scanning confocal microscope (Leica Microsystems, Heidelberg, GmbH), equipped with HCX PL APO ×63/1.4 and HCX PL APO ×100/1.4 oil immersion objectives, HyD detectors, and Leica Application Suite X software. HyVolution acquisition mode was used for quantification of mito-ncR-805 and Tom20 cellular co-localization with an optimized pixel size of 0.029 µm and step size of 0.136 µm. Mito-ncR-805-labeled structures were excited at 561 nm with fluorescence emission collected from 570 to 630 nm. Tom20-labeled structures were excited at 488 nm with fluorescence emission collected from 500 to 550 nm. Images were deconvolved with the Huygens Essential software (Scientific Volume Image, Hilversum, The Netherlands).

Plane projections, linescan analysis, 3D reconstructions, and final images were produced using MetaMorph (Molecular Devices Corporation). Linescan function of MetaMorph software was used to determine location of mito-ncR-805 relative to

Tom20 signal as described in Krokowski et al.[37]. Briefly, Linescans of 4 pixels wide were taken manually through the axis of mito-ncR-805 dots or throughout the nucleus. Afterwards, the fluorescence profile along the linescan was visualized for all three channels. Multiple (3–6) linescans were taken for each individual cell. 3D reconstruction was done using stacks containing all acquired planes for all three channels, and images were cropped to include only the selected regions of the original images.

*Chromogen in tissue detection of mito-ncR-805.* Paraffin-embedded parasagittal sections of inflated lungs were utilized for analysis. BaseScope Dultiplex Detection Assay (ACD, Inc.) was used according to the manufacturer's instructions, with minor modifications. Tissues were re-hydrated, per the manufacturer's instruction, treated with protease IV for 45 min at RT, then hybridized with BA-Mm-mt-D-loop-O1-1zz-C1 (builds amplification tree with Horse Reddish Peroxidase enzyme that results in chromogen developed as blue) and BA-Mm-Sftpb-3EJ-C2 (builds amplification tree with alkaline phosphatase enzyme that results in chromogen developed as red) probes, and amplification steps were performed following the manufacturer's instructions, except that AMP11 for green chromogen was 2 h at RT. Following hybridization and amplification, slides were incubated with Hoechst solution (1 mg/ml in water, Sigma-Aldrich) to counterstain nuclei and mounted on a microscope slide. Hoechst was used as a counter-staining to avoid the masking blue mito-ncR-805 signal using conventional Hematoxylin counter-staining. Leica DMLB microscope equipped with PL Fluotar, ×20, ×40, and ×100 Oil objectives (Leica Microsystems, Wetzlar, Germany), Rolera XR CCD camera (Q-Imaging, Surrey, Canada), and operated by the Q-Capture poro 7.0 software was used to acquire images. Images were acquired at ×20 and ×100 magnifications. Final images were produced using Adobe Photoshop CS5.1 (Adobe Systems Incorporated). Hoechst counterstain images were converted to the gray scale (copied from the blue channel of Adobe Photoshop) and overlaid on the original image with 38% opacity.

*Scoring lung tissues for mito-ncR-805 expression.* Scoring of lung tissues for mito-ncR-805 expression was done using modification as previously suggested for semi-quantitative microscopical evaluation of RNAScope, a 4-tier scoring system[47]; mito-ncR-805 dots were counted manually in 6–8 serendipitously chosen fields acquired at ×100 using the Adobe Photoshop CS5.1 software. For each field, the total number of cells in the field was determined as a number of Hoechst-visualized nuclei (T). The numbers of AETII cells per each field was determined as a number of Surfactant B-positive (red) cells (S). The number of non-AETII cells was determined by subtracting AETII cells number from the total cell number in the field (T − S). Than blue dots of mito-ncR-805 were counted per each individual Surfactant B-positive AETII cell (AETII dots). Each counted dot was color marked (red for AETII cell to avoid double counting). The average numbers of mito-ncR-805 dots per non-AETII cell was determined as the remaining number of dots divided by the number of non-AETII cells (non-AETII dots/(T − S)). For each animal, at least 25 AETII cells were scored (6–8 fields at ×100 will provide between 25 and 85 AETII cells depending on tissues inflation and degree of airspace enlargement) and >100 non-AETII cells. Numbers are specified for each condition. Two animals were analyzed per each condition. Score were assigned using the 4-tier scoring system: 0 —cells negative for mito-ncR-805, 1—few mito-ncR-805 spots in most cells, 2—moderate number of spots in all, and 3—high number of spots in all cells.

**Modified Gill lung tissue staining**. Slides were de-paraffinized by 3 xylene washes 3 min each; hydrated through series of alcohols, 3 min each; washed with water for 5 min, and stained with mixture of ½ Gills (Sigma GHS-332) and ½ Harris Hematoxylin (Sigma HHS-32) for the period of 16–24 h, to blacken all tissues. The stain was rinsed off using tap water; slides were washed with 0.02% Ammonium water for 5 min, followed by rinse with tap water, de-hydration, and mounting using Cytoseal-60 (Fisher Scientific 23-244-257). At least 10 random 1300 × 1030-pixel images of lung parenchyma stained with Modified Gills were acquired for each lung sample using a light microscope at ×20 magnification (Leica DMLB microscope equipped with Rolera XR CCD camera (Q-Imaging, Surrey, Canada) and operated by the Q-Capture poro 7.0 software). The Image J macro will account for debris and undesired holes in the tissue and automatically remove them from calculation of chord lengths (CLs). Large airways, blood vessels, and other non-alveolar structures were manually removed from the images using fill in option of the Adobe Photoshop. Images with removed non-alveolar tissues were saved in a separate for each analyzed animal files.

**Morphometric analysis of lung airspace**. Airspace was quantified by measuring a mean alveolar CL (the distance between alveolar walls, which is proportional to the amount of pulmonary emphysema) using a modification of a previously published automated image-processing algorithm[66,71,72]. ImageJ software (National Institutes of Health) equipped with a series of macros written to perform specific functions was used. The macro designed for morphometric measurement utilizes pre-drawn horizontal and vertical grids of one pixel width, overlays them onto the image, and then uses the image calculator function to convert airspaces into chord lines. The chord lines are then measured using the particle analysis function, which counts the number of pixels in every object on the image, since the image only includes lines of one pixel width, and this measurement serves as a length. This process is repeated for both the horizontal and vertical depictions of the image. The measurements are transferred to a macro excel spreadsheet that enables automatic calculation of descriptive statistics for that image. This process is completed as a loop for all of the images associated with a particular sample and descriptive statistics are also calculated for the batch of images as a whole resulting in the desired mean CL of the sample. The software then counted the number of lines that ended on or intercepted alveolar tissue. These data were used to calculate the average CL according to methods adapted from Dunhill[73].

**Nuclear extracts and nuclear RNA preparation**. The nuclear extracts were prepared by modification to Wang et al.[74]. Briefly, MLE12 cells, collected by trypsinization, were re-suspended in lysis buffer of 10 mM HEPES [pH 7.9], 10 mM KCl, 1.5 mM MgCl$_2$, and 0.5 mM dithiothreitol (DTT) supplemented with protease inhibitory tablet (Sigma-Aldrich Roche Biochemical Reagents), incubated 15 min on ice, followed by addition of nonidet P-40 to the final concentration of 0.325%. This was allowed to swell 10 min on ice with occasional shaking. Nuclear pellets were separated from the cytoplasm by centrifugation at 2500 rpm for 4 min at 4 °C. Nuclear pellets were washed once in the lysis buffer and re-suspended in 20 mM HEPES [pH 7.9], 0.45 M NaCl, 1 mM EDTA, and 0.5 mM DTT 1 protease inhibitor tablet (Complete mini EDTA free, Sigma-Aldrich Roche Biochemical Reagents) per 10 ml of lysis buffer. Nuclei were incubated with rocking at 4 °C for 30 min. Nuclear extracts were cleared by centrifugation at 12,000 rpm for 10 min.

Cytoplasm was further fractionated into cytosol and the rest using high-speed (100 K, at 4 °C, for 1 h) ultracentrifugation. Each supernatant was divided into two: half for western blot analysis, and the second half for RNA extraction and analysis.

For RNase treatment experiments, nuclear pellets, re-suspended in lysis buffer supplemented with 0.3 M Sorbitol, were treated with RNase I (Ambion, AM2294, 100 U/100 μl), at 4 °C, for 1 h, to eliminate cytosolic RNAs, and suspended in Trizol for further RNA analysis.

**Transfection procedure**. MLE12 cells were seeded at the density of 50,000 cells/ml on polystyrene plates with RPMI Complete media and left to adhere for 12 h. Media was removed and replaced with OptiMEM media. After 10–12 h, cells were transfected with siPORT NeoFX (Life Technologies, Thermo Fisher Scientific) transfection reagent supplemented with either 25 nM AI805 (AM11866) for downregulation studies, 25 nM Pre-miR Negative Control (AM17110), or 10 nM ncR805 for overexpression studies. ncR805 is a synthetic 70-bp mito-ncR-805, stabilized by modification and additions to 5′ and 3′ termini RNA oligo, where "m" stands for 2-O-methyl-RNA; and "*" stands for phosphorothioate:

mG*mA*mA*UUGAUCAGGACAUAGGGUUUGAUAGUUAAUA UUAUATGUCUUUCAAGUUCUUAGUGUUUUUGGG*mG*mU*mU. After 6 h, transfection complex was removed, and cells were incubated in their regular medium.

**Mitochondrial gene expression analysis**. RNA from each sample was cDNA synthesized using the SuperScript III First-Strand Synthesis SuperMix (Invitrogen). The relative quantity of specific RNAs was measured by qRT-PCR using the VeriQuest SYBR Green qPCR Master Mix (Affymetrix) with the StepOnePlus Real-Time PCR System (Applied Biosystems). Mitochondrial primers were:

| Gene name | Amplicon size | Forward primer | Reverse primer |
|---|---|---|---|
| ND6 mouse | 138 | 1mmND6_for TCTTGATGGTTTGG GAGATTGGT | 1mmND6_rev CCCGCAAACAAAG ATCACCC |
| tRNA-Pro | 186 | 1mttRNAPro_for GTGGGGAGTAGCTCC TTCTTC | 1mttRNAPro_rev GGCCAACTAGCCT CCATCTC |
| ND1 mouse | 134 | 1ND1_for TCCGAGCATCTTATCC ACGC | 1ND1_rev GTATGGTGGTACT CCCGCTG |
| COX1 mouse | 145 | 1Cox1_for TCGGAGCCCCAGATA TAGCA | 1Cox1_rev TTTCCGGCTAGAG GTGGGTA |
| rRNA12S tRNAval | 155 | 112StRNAval_for CCCGTCACCCTCCTCA AATTA | 112StRNAval_rev GGGTGTAGGCCAG ATGCTTT |
| rRNA16S | 186 | 1_tRNAval16S_for AAAGCATCTGGCCTAC ACCC | 1_tRNAcal16S_rev TCATCTTTCCCTTG CGGTACT |
| 18s rRNA | 131 | TTGACGGAAGGGCACC ACCAG | GCACCACCACCCA CGGAATCG |

**Mitochondrial protein synthesis**. Metabolic labeling of mtDNA-encoded proteins was performed as described[75]. Briefly, 1 h prior to the end of the treatment, cells' media was changed to MLE media with dialyzed FBS. Five minutes prior to

labeling, 100 µg/ml emetine (Sigma) to inhibit cytoplasmic protein synthesis or combination of 100 µg/ml emetine and 40 µg/ml chloramphenicol to inhibit both cytoplasmic and mitochondrial protein synthesis were added, followed by 1 h incubation in the presence of 200 µCi/ml of EasyTag™ EXPRESS 35S Protein Labeling Mix, (Perkin-Elmer). Labeling media was exchanged with regular media containing respective protein synthesis inhibitor, and incubation was continued for additional 10 min. Cells were washed with PBS, scraped, pelleted by centrifugation at 6000 rpm for 5 min, and lysed by 10-min incubation on ice in PBS containing 2% (w/v) lauryl maltoside. Insoluble material was removed by centrifugation at $10,000 \times g$ for 10 min. Protein concentrations were determined using the Bradford protein assay (Bio-Rad). Fifty micrograms of protein were separated on 15% Tris-glycine gels. The amounts of 35S-labeled mitochondrial proteins were visualized using a PhosphorImager. The rates of mitochondrial translation were determined from two independent protein-labeling experiments.

**Pathway-focused gene expression analysis.** Pathway-focused gene expression analysis was performed using RT² Profiler PCR Arrays of Mouse Mitochondria plates and Mouse Mitochondrial Energy Metabolism plates (Qiagen, PAMM-087Z and PAMM-008Z) according to the manufacturer's instruction. All Qiagen RT² Profiler plates qRT-PCRs were performed using RT² SYBR Green ROX qPCR Mastermix (330520) on StepOne Plus Real-Time PCR system; Applied Biosystems). Results were processed by the Qiagen algorithm software available at the data analysis web portal at http://www.qiagen.com/geneglobe. Samples were assigned to controls and test groups. CT values were normalized based on a Qiagen manual algorithm selection with GAPDH as a reference gene.

**Metabolic labeling of MLE12 cells.** MLE12 cells were transfected with either AI805 or ncR805 and non-targeting sequence; 30 h post-transfection, cells were exposed to 20 mM [U-$^{13}$C]glucose for 12 h, harvested, and metabolic flux was determined following $^{13}$C label incorporation into various metabolites. After incubation, 0.5 ml of the removed media was saved from one well of each condition and each plate. Cells were washed with ice-cold saline (3 times), fixed and collected by adding 1 ml of 80% ethanol cooled on dry ice water bath, and either processed immediately or stored at −80 °C for further analysis.

**Assay of media [U-$^{13}$C]glucose enrichment.** Glucose isotopic enrichment was determined following Gao et al.[76] with modifications. Briefly, glucose was extracted by the addition of 500 µl of ice-cold ethanol to 50 µl of media. Samples were mixed and incubated on ice for 30 min. Samples were centrifuged at 4 °C for 10 min at 14,000 rpm and ethanol was transferred to gas chromatography/mass spectrometry (GC/MS) vial and evaporated to dryness in SpeedVac evaporator. Glucose was converted to its pentaacetate derivative by the reaction with 150 µl of acetic anhydride in pyridine (2:1, vol/vol) at 60 °C for 30 min. Sample was evaporated to dryness and glucose derivative was reconstituted in 80 µl of ethyl acetate and transferred to GC/MS insert. Samples were injected in duplicate and masses 331–337 containing M0–M5 isotopomers were monitored. Enrichment was determined as a ratio of $(M5)/(\Sigma_{M0-M5})$.

**Metabolite extraction.** Metabolite extraction was done as described[77]. Briefly, cell extract was vortexed and sonicated using ultrasonicator for 30 s on, 30 s off, alternating for 10 min. Cells were pelleted by centrifugation at 4 °C for 10 min at 14,000 rpm. Supernatant was transferred to GC/MS vials and evaporated to dryness under gentle stream of nitrogen. Keto and aldehyde groups were reduced by addition of 10 µl of 1 N NaOH and 15 µl NaB$^2$H$_4$ (prepared as 10 mg/ml in 50 mM NaOH). After mixing, samples were incubated at RT for 1 h and then acidified by 55 µl of 1 N HCl by dropping the acid slowly and evaporated to dryness. Fifty microliters of methanol were added to precipitate boric acid. Internal standard was added (10 µl of 17:0 FA, 0.1 mg/ml). Samples were evaporated to dryness and reacted with 40 µl of pyridine and 60 µl of tert-butylbis(dimethylsilyl) tri-fluoroacetamide with 10% trimethylchlorosilane (Regisil) TBDMS at 60 °C for 1 h. Resulting TBDMS derivatives were injected into GC/MS.

**GC/MS conditions.** Analyses were carried out on an Agilent 5973 mass spectrometer equipped with 6890 Gas Chromatograph. A HP-5MS capillary column (60 m × 0.25 mm × 0.25 µm) was used in all assays with a helium flow of 1 ml/min. Samples were analyzed in Selected Ion Monitoring mode using electron impact ionization. Ion dwell time was set to 10 ms. The following metabolites were detected: α-ketoglutarate, alanine, aspartate, citrate, fumarate, glutamate, lactate, malate, oxaloacetate, pyruvate, serine, and succinate. The acetyl-CoA levels were not directly measured because acetyl-CoA originates from the two sources, glucose and fatty acid oxidation. In addition, the acetyl-CoA pool turns over very fast and its size is very small.

Citrate/acetyl-CoA refers to the acetyl-CoA moiety of citrate. Similarly, the marking citrate/OAA refers to the OAA moiety of citrate.

*Calculations.* Fractional metabolic flux was determined by applying precursor ([U-$^{13}$C]glucose) to product ([$^{13}$C]-labeled metabolites) relationship. Molar percent enrichment (MPE) of metabolites were determined in the same manner as for glucose (see above). Fractional metabolic flux was calculated as follows: $\text{MPE}_{\text{product}}/\text{MPE}_{\text{precursor}}$. Absolute metabolic rate shown in Fig. 7a–c was determined as a product of fractional metabolic flux and pool size, i.e., relative concentration of metabolite of interest.

**Mitochondrial bioenergetics measurements.** MLE12 cells were seeded at 10,000/well (80 µl), transfected with either Negative Control or AI805 as described above. Forty hours following transfection and 1 h prior to the measurements, RPMI complete was changed to unbuffered DMEM medium (Seahorse Bioscience, USA) supplemented with 1% FBS, 4 mM glutamine (Sigma Aldrich, USA), 2 mM sodium pyruvate (Gibco, USA), 10 mM glucose (Sigma Aldrich, USA), and 100 µM Insulin (Sigma Aldrich, USA), and mitochondrial bioenergetics was measured using a Seahorse XF$^e$96 extracellular flux analyzer and the XF Cell Mito Stress Kit (Seahorse Bioscience, USA) as described before[78]. Real-time measurement of OCR was obtained as per the manufacturer's instructions by sequential treatment with 1 µM oligomycin (Sigma O4876), 1 µM *p*-triflouromethoxyphenylhydrazone (FCCP, Sigma C2920), and a 2 µM mixture of rotenone/antimycin A (Sigma R8875 and A8674) and normalized to total protein amount measured by the Pierce BCA Protein Assay Kit (Thermo Scientific, USA). Basal respiration was calculated as average of first 3 readings before oligomycin addition − average of last 3 readings after rotetone addition. Proton leak was calculated as average of 3 readings after oligomycin addition − average of last 3 readings. ATP production was calculated as basal respiration − proton leak. Maximal respiration was calculated as average of 3 reading after FCCP stimulation − average of last 3 readings. Spare respiratory capacity was calculated as maximal respiration − basal respiration. Coupling efficiency was calculated as 100 × ATP production/non-mitochondrial respiration, where non-mitochondrial respiration is the average of the last 3 readings after rotetone and antimycin A addition (Supplementary Data).

**Mitochondrial potential assays.** Mitochondrial membrane potential was measured using the JC-10 Mitochondrial Membrane Potential Assay Kit (ab112134, Abcam, USA) according to the manufacturer's instructions. Results were read by a Biotek Synergy HT spectrophotometer.

**Scratch wound healing assay.** Scratch wound healing assay was performed as previously described[79]. Briefly, MLE12 cells were grown to 80% confluence, scratched using a p10 pipette tip, and imaged at the time of scratch and every 12 h afterwards.

**Statistics and reproducibility.** Data are expressed as means of at least 3 independent experiments ± SD. For each experiment, the precise number of independent biological experiments performed on different dates and the number of biological replicates in each experiment are specified in figure legends, Source data are provided in Supplementary Data. Fold change for qPCR was determined using the $2^{-\Delta\Delta Ct}$ method[80,81]. Student's *t* test was used to determine the *P* value. Values of $P \leq 0.05$ were considered statistically significant.

**Reporting summary.** Further information on experimental design and research design is available in the Nature Research Reporting Summary linked to this article.

## Data availability

The datasets generated and analyzed during the current study are available at GEO: GSE131188 (microarray dataset) and GSE131439 (RNAseq dataset). Source data and calculations for all experiments can be found in Supplementary Data. Uncropped images of western and Northern blots are provided in Supplementary Fig. 5.

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

## Acknowledgements

We would like to thank Dr. Gerald Shadel, Salk Institute for Biological Studies for help with the design of mitochondria and Ago2-related experiments and critical reading and editing of the manuscript; Linda Gonzales, Children's Hospital of Philadelphia and Dr. Torry A. Tucker, University of Texas Health Sciences Center for their help with isolation of AETII cells; Dr. Steven Shapiro, University of Pittsburgh for help and consultation with morphomery measurments; Jeremy A. Brown for adaptation of lung morphometric algorithm to measure alveolar airspace for ImageJ software; Dr. Judish Drazba, Lerner Research Institute Imaging Core Director, Cleveland Clinic for her help with imaging; Dr. Norma Nowak and Jonathan Bard, SUNY at Buffalo for help with RNAseq analysis; and Dr. Kenneth Farabaugh for editorial help. This project was supported by funding from FAMRI Young Clinical Scientist Award to A.B.-P. (Grant #092207), the Gainard Golz research fund to Y.P., National Institutes of Health Grants R37-DK060596 and R01-DK053307 to M.H., R01HL127349, R01 HL118536, and U01 HL108642 to N.K., R01 GM108807 to S.H.G., R01 HL130283 and AG053495 to M.-J.K., and National Science Centre, Poland Grant 2018/30/E/NZ1/00605 to D.K., and American Cancer Society Institutional Research Grant #IRG-19-137-20 to J.Z.

## Author contributions

A.B.-P. designed and supervised the project, performed multiple experiments including ISH and FISH-IF development, built the figures, and wrote the manuscript with input and edits from other authors. R.J. constructed RNAseq library and performed Northern blot and mitochondrial translation analyses. A.J.D. measured levels of mito-ncR-805 in cells and selected tissues, performed pathway analysis, and contributed to initial writing. K.P. performed the initial screen, which was analyzed by K.P. and H.K. I.B. and P.E. performed metabolomics analysis. Z.-W.Y., J.Z., and D.M.T. measured bioenergetics of cells. B.J.G. and D.K. performed mtDNA copy number measurements and mitochondrial gene expression analysis. N.A.P. and J.W. performed western blot analysis. B.J.G. and J.W. performed shRNA depletion. N.A.P. performed the cell culture part of the experiments for metabolomics, fractionation, Northern blot analysis, and RNAseq library. J.W. performed cell culture part of mitochondrial translation experiments; N.D.M. and A.M. performed deep sequencing and analyzed results. C.E.N. exposed mice to smoke, collected tissues, and participated in the validation of the initial screen. S.H.G. and F.A. advised and helped in the isolation of primary AETII cells. S.H.G. critically read and edited the manuscript. P.A.L. advised on AETII experiments and critically read and edited the manuscript. M.-J.K. provided mice for the isolation of primary AETII cells and critically read and edited the manuscript. N.K. participated in the design of the project, performed and analyzed the initial screen, and critically read and edited the manuscript. Y.P. participated in study design and mice and AETII experiments and contributed to writing. M.H. participated in the design and supervision over the whole project and critically read and edited the manuscript.

## Competing interests

The authors declare no competing interests.
