## [Peer Review File · Communications Biology]

This manuscript has been previously reviewed at another Nature Research journal. This document only contains reviewer comments and rebuttal letters for versions considered at Communications Biology.

REVIEWERS' COMMENTS:

Reviewer #1 (Remarks to the Author):

Comments on the revision of the manuscript "Retrograde signaling by mtDNA-encoded non-coding RNA preserves mitochondrial function" by Blumental-Perry et al.

The authors have done an excellent job with the revision of the manuscript. They have responded to the comments of reviewer 1 and 2 appropriately. The manuscript has become more stringent and has greatly improved upon revision and by concentrating on chronic stress responses mediated by CSE and omission of the tumor data. In addition, the authors have now focused on the long 70nc form of the ncRNA805 and omitted their data on the short form, which had raised several concerns by the reviewers. Moreover, development of FISH detection of the ncRNA805 now provides improved insight into the subcellular localization of this ncRNA as well as its tissue distribution. The new Figure 2 gives important evidence for the mitochondrial origin of the ncRNA using multiple experimental approaches.

Minor concerns:

1. The use of the term "smoker" in Fig. 4 is misleading and could be understood as human smokers. Mice are not smokers but they are mice exposed to smoke. The name should be changed accordingly.

2. The authors should be more precise in drawing their conclusions and in their wording: e.g page 7:

"Together, our data indicate that miR-805 is induced in response to CSE in both MLE12 and primary AETII cells ex and in vivo." and "The levels of miR-805 were down-regulated in all lung samples tested (Supp. Fig. 1D-G)." lead to the following conclusion: "We concluded that CS exposure of mice or CSE exposure of cells induces miR-805 specifically in AETII cells in lungs." This is far fetched and cannot be seen in the data presented at that point. It might be indicated but can only be drawn from some data later in Fig 4 when ISH is shown for lung tissue of smoke-exposed mice.

The same is true for the statement: "Therefore, expression of miR-805 increases in response to CS exposure of mice in secretory cells and in local...". The data you show only refer to liver and AETII cells and not to secretory cells in general. You can speculate that this is specific for secretory cells, but then you have to make that clear in the text and not describe it as facts.

We are very happy that Communication Biology is interested in publishing our findings presented in the manuscript entitled "Retrograde signaling by a mtDNA-encoded noncoding RNA preserves mitochondrial bioenergetics"

We have addressed all the remaining reviewers' comments:

Concern:

1. The use of the term "smoker" in Fig. 4 is misleading and could be understood as human smokers. Mice are not smokers but they are mice exposed to smoke. The name should be changed accordingly.

Response:

The name was changed in the old Fig. 4 (new Fig. 5) according to the reviewers' suggestion.

Concern:

2. The authors should be more precise in drawing their conclusions and in their wording: e.g page 7: "Together, our data indicate that miR-805 is induced in response to CSE in both MLE12 and primary AETII cells ex and in vivo." and "The levels of miR805 were down-regulated in all lung samples tested (Supp. Fig. 1D-G)." lead to the following conclusion: "We concluded that CS exposure of mice or CSE exposure of cells induces miR-805 specifically in AETII cells in lungs." This is far fetched and cannot be seen in the data presented at that point. It might be indicated but can only be drawn from some data later in Fig 4 when ISH is shown for lung tissue of smoke-exposed mice. The same is true for the statement: "Therefore, expression of miR-805 increases in response to CS exposure of mice in secretory cells and in local...". The data you show only refer to liver and AETII cells and not to secretory cells in general. You can speculate that this is specific for secretory cells, but then you have to make that clear in the text and not describe it as facts.

Response:

The conclusions were modified according to the reviewers' suggestions. The paragraph now reads as:

"We tested whether induction of miR-805 is a general response of different cell types. miR-805 levels were compared in total lung and liver lysates of control and CS-exposed mice. The levels of miR-805 were downregulated in total lung CS-exposed samples (Supp. Fig. 1d-e). Liver is a tissue that shares common properties with AETII cells: secretory cells with strong reparative abilities. Expression of miR-805 was elevated in the livers of CS-exposed mice (Supp. Fig. 1f). Therefore, increase in miR-805 expression in response to CS-exposure in mice maybe specific to secretory and local niche progenitor cells"

In addition, we changed all graphs to be presented in the format required by CB. All original data sources were summarized in Supplementary Data that has 10 different sheets (legend is provided below), containing all micro-array data, qPCR data, counting data, morphometry measurements data, as well as previously provided qPCR pathfinder array data, metabolomics and seahorse analysis and calculations data. Green/red colors were avoided in schemes and images.